# Differentially Private Quantiles with Smaller Error

**Jacob Imola**
BARC, University of Copenhagen
Denmark
jaim@di.ku.dk

**Fabrizio Boninsegna**
University of Padova
Italy
fabrizio.boninsegna@phd.unipd.it

**Hannah Keller**
Aarhus University
Denmark
hkeller@cs.au.dk

**Anders Aamand**
University of Copenhagen
Denmark
aa@di.ku.dk

**Amrita Roy Chowdhury**
University of Michigan, Ann Arbor
United States of America
aroyc@umich.edu

**Rasmus Pagh**
BARC, University of Copenhagen
Denmark
pagh@di.ku.dk

## Abstract

In the approximate quantiles problem, the goal is to output $m$ quantile estimates, the ranks of which are as close as possible to $m$ given quantiles $0 \leq q_1 \leq \cdots \leq q_m \leq 1$. We present a mechanism for approximate quantiles that satisfies $\varepsilon$-differential privacy for a dataset of $n$ real numbers where the ratio between the distance between the closest pair of points and the size of the domain is bounded by $\psi$. As long as the minimum gap between quantiles is sufficiently large, $|q_i - q_{i-1}| \geq \Omega\left(\frac{m \log(m) \log(\psi)}{n\varepsilon}\right)$ for all $i$, the maximum rank error of our mechanism is $O\left(\frac{\log(\psi) + \log^2(m)}{\varepsilon}\right)$ with high probability. Previously, the best known algorithm under pure DP was due to Kaplan, Schnapp, and Stemmer (ICML '22), who achieved a bound of $O\left(\frac{\log(\psi) \log^2(m) + \log^3(m)}{\varepsilon}\right)$. Our improvement stems from the use of continual counting techniques which allows the quantiles to be randomized in a correlated manner. We also present an $(\varepsilon, \delta)$-differentially private mechanism that relaxes the gap assumption without affecting the error bound, improving on existing methods when $\delta$ is sufficiently close to zero. We provide experimental evaluation which confirms that our mechanism performs favorably compared to prior work in practice, in particular when the number of quantiles $m$ is large.

## 1 Introduction

Quantiles are a fundamental statistic of distributions with broad applications in data analysis. In this paper, we consider the estimation of *multiple* quantiles under differential privacy. Given a dataset $X$ of $n$ real numbers and quantiles $0 < q_1 < \cdots < q_m < 1$ the goal is to output estimates $z_1, \ldots, z_m$ such that the fraction of data points less than $z_i$ is approximately $q_i$. We measure the error by the difference between the rank of $z_i$ in $X$ and the optimal rank $q_i n$. For the ease of exposition, we first consider the case where elements of $X$ are integers in $\{1, \ldots, b\}$ and the error probability is bounded by $1/b$. In particular, the ratio $\psi$ between the closest pair of points and the domain size is bounded

39th Conference on Neural Information Processing Systems (NeurIPS 2025).

by $b$. Past mechanisms fall into two categories based on the type of privacy guarantee achieved. For pure $\varepsilon$-differential privacy, the best known bound on maximum rank error of $O\left(\log(b)\log^2(m)/\varepsilon\right)$ is due to Kaplan, Schnapp, and Stemmer [18]. On the other hand, work on approximate differential privacy has largely focused on controlling how the error grows with the domain size $b$ [8, 3, 17, 11]. These results reduce the dependence on $b$ down to $\log^*(b)$, but introduce $\log(\frac{1}{\delta})$ factors—for small values of $\delta$, for example when $\log(1/\delta) > \log(b)\log^2(m)$, the bound on rank error exceeds that of methods guaranteeing pure differential privacy.

Our goal is two-fold: to improve the rank error of private quantile estimates *both* under pure differential privacy, and under $(\varepsilon, \delta)$-differential privacy with small values of $\delta$. To this end, we make the following contributions.

- We present a mechanism that satisfies $\varepsilon$-differential privacy for any quantiles satisfying a mild gap assumption: specifically, that $|q_i - q_{i-1}| \geq \Omega(\frac{m\log(m)\log(b)}{n\varepsilon})$ for all $i$. This condition depends only on the (public) queried quantiles—if they do not meet this assumption the protocol can be safely halted without accessing any private data. For quantiles meeting the assumption, our mechanism achieves a maximum rank error of $O\left(\frac{\log(b)+\log^2(m)}{\varepsilon}\right)$ with high probability, saving a factor $\Omega(\min(\log(b), \log^2(m)))$ over past purely private mechanisms.

- We also present an $(\varepsilon, \delta)$-differentially private mechanism that relaxes the gap assumption to be independent of $b$ without affecting the error bound. Notably, our error guarantee remains *free* of any dependence on $\delta$; the parameter $\delta$ only appears in the assumption on the gap between quantiles.

For both the mechanisms, our improvement stems from the use of continual counting techniques to randomize the quantiles in a correlated manner. We provide an experimental evaluation on real-world datasets which validates our theoretical results. The improvement is most pronounced when the number of quantiles $m$ is large, particularly under the substitute adjacency. In this setting, our mechanism improves the accuracy compared to [18] by a factor of 2 when estimating 200 quantiles with $n = 500,000$, $\varepsilon = 1$, and $\delta = 10^{-16}$.

## 1.1 Relation to Past Work

The problem of quantile estimation is closely related to the problem of learning cumulative distributions (CDFs) and threshold functions [14, 7]. Learning of threshold functions is usually studied in the *statistical* setting where data is sampled i.i.d. from some real-valued distribution. For *worst-case* distributions the problem has sample complexity that grows with the support size, so in particular we need to assume that the support is finite or that the distribution can be (privately) discretized without introducing too much error. Feldman and Xiao [14] established a sample complexity lower bound of $\Omega(\log b)$ for the quantile estimation problem under pure differential privacy. Bun, Nissim, Stemmer, and Vadhan [7] demonstrated a lower bound of $\Omega(\log^* b)$ for the same task under $(\varepsilon, \delta)$-differential privacy, and a mechanism with nearly matching dependence on $b$ was developed in a series of papers [8, 3, 17, 11]. We note that these results focus on $b$ and do not have an optimal dependence on the privacy parameter $\delta$. For example, the algorithm of Cohen, Lyu, Nelson, Sarlós, and Stemmer [11] has sample complexity (and error) proportional to $\tilde{O}(\log^* b)$, which is optimal, but this bound is multiplied by $\log^2(1/\delta)$. For extremely large data domains, [11] can therefore outperform our algorithm; however, for domain sizes encountered in practice, the higher dependence on $\delta$ will likely outweigh this improvement. Our mechanism for approximate DP yields error independent of $\delta$ and therefore outperforms existing mechanisms when $\delta$ is small. With a slightly worse dependence on $b$, Kaplan, Ligett, Mansour, Naor, and Stemmer [17] achieved an error proportional to $\log(1/\delta)$. Earlier work [8, 3] had a weaker dependence on $b$ and $\delta$.

The problem of estimating $m$ quantiles under *pure* differential privacy has been explored by Gillenwater, Joseph, and Kulesza [15] as well as Kaplan, Schnapp, and Stemmer [18]. The latter proposed an algorithm with error $O(\frac{\log^2(m)(\log(b)+\log(m))}{\varepsilon})$. In the *uniform quantile* setting, where quantiles are evenly spaced, they improved this bound by a factor $O(\log m)$. Their approach is inspired by the work of Bun, Nissim, Stemmer, and Vadhan [7], solving the problem of single quantiles using the exponential mechanism instead of an interior point algorithm. The problem is solved recursively by approximating the middle quantile $q_{m/2}$ and recursing on the dataset relevant for the first and

| Privacy Guarantee | Minimum Gap | Error | Notes |
|---|---|---|---|
| $(\varepsilon, 0)$-Differential Privacy, Corollary 5.2 | $\Omega\left(\frac{m\log(m)\log(b)}{\varepsilon n}\right)$ | $O\left(\frac{\log(b)+\log^2(m)}{\varepsilon}\right)$ | Saves a $\Omega(\min(\log(b), \log^2(m)))$ factor over [18]. |
| $(\varepsilon, \delta)$-Differential Privacy, Corollary 5.3 | $\Omega\left(\frac{\log(m)\log(m/\delta)+\log(b)}{\varepsilon n}\right)$ | $O\left(\frac{\log(b)+\log^2(m)}{\varepsilon}\right)$ | Error independent of $\delta$ unlike prior work [11, 17]. |

Table 1: Summary of our theoretical results. We consider both add/remove and substitute adjacency. Our privacy analysis is tighter for the latter.

second half of the quantiles, respectively. If the quantiles satisfy our maximum gap assumption, then our algorithm enjoys lower error by a $\min\{\log^2(m), \log(b)\}$ factor when $b \geq m$. If one is willing to relax to approximate DP, we can significantly reduce the gap assumption for the same error. By combining quantiles, the gap assumption can be eliminated entirely, and the error will be $O(\frac{1}{\varepsilon}(\log(b) + \log(m)\log\frac{m}{\delta}))$; this is still less than [18] when $\frac{1}{\delta} \ll b$, usually the case for larger data domains. The properties of the algorithm of [18] in the statistical setting was investigated by Lalanne, Garivier, and Gribonval [19]. They also considered an algorithm based on randomized quantiles, but it relies on strong assumptions on the smoothness of the distribution.

Differentially private quantiles has received attention under different problem formulations. Some work takes the error function to be the absolute difference between the estimate and the true quantile [12, 22], instead of the rank error. This gives rise to a fundamentally different problem, and distribution assumptions are typically needed to ensure good utility. The problem has also been considered in streaming [2] and under local differential privacy [12, 1], though the different natures of these problems prevents the techniques from carrying over.

**Our Approach.** Similar to [18] we also solve the problem by splitting it into $m$ subproblems, referred to as "slices". Each slice is a contiguous subsequences of the sorted input data $X = \{x_i\}_{i=1,\dots,n}$; that is, each slice must consist of the elements $x_{(i)}, \dots, x_{(j)}$ for two indices $i < j$. However, instead of using divide-and-conquer, we take a different approach – we propose a way to choose random slices around the quantiles using techniques from continual counting [13]. Assuming the quantiles are sufficiently spaced, each quantile can be approximated by applying the exponential mechanism to a subset of points around the quantile. The total mechanism then consists of two steps: (1) splitting the dataset into disjoint subsets around the quantiles using private continual counting, and (2) applying the exponential mechanism to each disjoint subproblem. The main technical challenge is that modifying one data point can modify the data contained in many of the subsets. To circumvent this issue, we must add correlated noise to each quantile before forming the subsets, which has the effect of hiding the modifications created in all the slices by the change in a single data point. Our privacy analysis introduces a novel mapping for adjacent datasets $X$ and $X'$: by carefully aligning the noise introduced via continual counting, we ensure that at least $m - 1$ slices remain identical. This gives approximate differential privacy (since the mapping is not exact), but the resulting $\delta$ parameter can be made extremely small given sufficient spacing between quantiles. We can then achieve pure differential privacy by mixing in a uniformly random output with a very small probability. Although slicing has also been used in prior work [11], we are the first to apply continual counting in this context and achieve utility guarantees that are independent of $\delta$.

**Limitations.** Our algorithms introduce a quantile gap assumption not present in prior work. However, as long as the number of quantiles is not too large, this assumption is often met—data analysts often care about a limited number of summary statistics (e.g., $10\%, 20\%, \dots, 90\%$). A particularly important case is when the quantiles are equally spaced, which is closely tied to the problem of CDF estimation. For approximate DP, our gap assumption is milder, at $O(\frac{1}{n\varepsilon}(\log(b) + \log(m)\log\frac{m}{\delta}))$—this is usually significantly less than 1 in practice, and for the realistic parameters tested in our experiments, the required gap was $< 0.005$. However, other techniques may be preferable when the quantiles are closer than the gap. We believe that a tighter analysis may relax the gap assumptions.

## 2 Background

Our setup is identical to that of Kaplan Schnapp, and Stemmer [18]. That is, we consider a dataset $X = \{x_i\}_{i=1,\dots,n}$ where $x_i \in [a, b] \subset \mathbb{R}$. We say a dataset $X$ has minimum separation $g$ if

$\min_{i \neq j} |x_i - x_j| \geq g$[1]. Unless explicitly specified, without loss of generality we assume that $a = 0$, $g = 1$, since any general case can be reduced to this setting via a linear transformation of the input. Our results can be translated to the general case by replacing $b$ with $\psi = \frac{b-a}{g}$. Given a set $Q$ of $m$ quantiles $0 \leq q_1 < \cdots < q_m \leq 1$, the quantile estimation problem is to privately identify $Z = (z_1, \ldots, z_m) \in [0, b]^m$ such that for every $i \in [m]$ we have $\mathsf{rank}_X(z_i) \approx \lfloor q_i n \rfloor$. We consider the following error metric:

$$\mathrm{Err}_X(Q, Z) = \max_{i \in [m]} |\mathsf{rank}_X(z_i) - \lfloor q_i n \rfloor|$$

This error metric has an intuitive interpretation as the difference between the estimate's rank and the desired rank, and is the one more often considered in the DP literature [18, 15, 7]. For convenience, we let $r_i = \lfloor q_i n \rfloor$ denote the *rank* associated with each quantile. We denote by $x_{(i)}$ the $i$th smallest element of $X$ (i.e., the sorted order of the dataset).

**Differential Privacy.** We consider two notions of adjacency in our privacy setup. Two datasets $X, X'$ are add/remove adjacent if $X' = X \cup \{x\}$ or $X = X' \cup \{x\}$ for a point $x$. We say $X, X'$ are substitute adjacent if $|X| = |X'|$, and $|X \triangle X'| \leq 2$. Thus, $X'$ may be obtained from $X$ by changing one point from $X$. Differential privacy is then defined as:

**Definition 2.1.** *A mechanism $\mathcal{M}(X) : [0, b]^n \to \mathcal{Y}$ satisfies $(\varepsilon, \delta)$-differential privacy under the add/remove (resp. substitute) adjacency if for all datasets $X, X'$ which are add/remove (resp. substitute) adjacent and all $S \subseteq \mathcal{Y}$, we have*

$$\Pr[\mathcal{M}(X) \in S] \leq e^\varepsilon \Pr[\mathcal{M}(X') \in S] + \delta.$$

In our results, we explicitly specify which adjacency definition is used. While the two notions are equivalent up to a factor of 2 in privacy parameters, our bounds for substitute adjacency are slightly tighter and not directly implied by those for add/remove adjacency.

## 3 Proposed Algorithm: SliceQuantiles

We begin with an intuitive overview of our algorithm, followed by the full technical details. For ease of exposition, we focus here on add/remove adjacency and defer substitution adjacency to Section 4.

### 3.1 Technical Overview

At a high level, our private quantiles algorithm SliceQuantiles applies a private single-quantile algorithm SingleQuantile to *slices* of the input dataset, which are contiguous subsequences of the sorted dataset. To ensure the accuracy of the overall scheme, they must meet the two conditions:

- Each slice must be sufficiently large because the accuracy of SingleQuantile is typically meaningful only when there is enough input data.
- The slices must be centered around the desired rank $r_i$, i.e. consist of data with rank approximately $r_i$.

Following the criteria outlined above, one might attempt to use slices $S_1, \ldots, S_m$, where each slice is defined as $S_i = x_{(r_i - h)}, \ldots, x_{(r_i + h)}$ for some sufficiently large integer $h$ [2], and estimate quantile $q_i$ by applying SingleQuantile to $S_i$. However, as noted in prior work [11, 7], this does *not* have a satisfying privacy parameter. Consider an adjacent dataset $X'$, where a new point with rank $s$, $x'_{(s)}$, is added. This produces a change in each slice $S'_t, \ldots, S'_m$, where $t$ is the smallest index such that $s \leq r_t - h$. The naive approach would be to use composition to analyze the quantile release, causing the error to increase by a factor of $\mathsf{poly}(m)$.

Instead, we make a more fine-grained analysis based on the following observation: for $i > t$ the slice pairs $\{S_i, S'_i\}$ are shifted by exactly 1, i.e., $S_i = x'_{(r_i - h + 1)}, \ldots, x'_{(r_i + h + 1)}$. Our goal is to hide

---

[1]This is easy to enforce, as a minimum separation can always be created by adding a small amount of noise to each data point (or by adding $\frac{i}{n}$ to point $i$). Importantly, if the data is not separated, it affects *only* the utility guarantees of our algorithms—not their privacy guarantees.

[2]For now, assume the slices do not overlap — we address this issue later.

---

**Algorithm 1** SliceQuantiles

---

1: **Input:** $X, r_1, \ldots, r_m, (w, \varepsilon_1), (\ell, \varepsilon_2), \gamma, [0, b]$
2: Set $h = \left\lceil \frac{\ell-1}{2} \right\rceil$
3: **Require**: $r_1, \ldots, r_m \in \mathsf{Good}_{m,n,w+h}$
4: $\tilde{r}_1, \ldots, \tilde{r}_m \leftarrow \mathsf{CC}_{\varepsilon_1}(r_1, \ldots, r_m)$            ▷ Post-process the noisy ranks to integers.
5: Flip a coin $c$ with heads probability $\gamma$
6: **if** $c$ is heads or $\tilde{\mathbf{r}} \notin \mathsf{Good}_{m,n,h}$ **then**
7:      Sample $z_1, \ldots, z_m$ i.i.d. from $[0, b]$
8:      **return** $z_1, \ldots, z_m$
9: **end if**
10: **for** $i = 1$ to $m$ **do**
11:      $S_i \leftarrow \left[ x_{(\tilde{r}_i - h)}, \ldots, x_{(\tilde{r}_i + h)} \right]$            ▷ Get the perturbed slice
12:      $z_i \leftarrow \mathsf{SingleQuantile}_{\varepsilon_2, [0,b]}(S_i)$
13: **end for**
14: **return** $z_1, \ldots, z_m$

---

this by randomly perturbing the ranks to noisy values $\tilde{r}_i$ such that it is nearly as likely to observe $\tilde{r}_i = v_i + e_i$, for any possible integers $v_1, \ldots, v_m$, and any "shifting" values $e_1, \ldots, e_m$ satisfying

$$e_i = \begin{cases} 0 & i \leq t \\ c & i > t \end{cases} \tag{1}$$

for a given index $0 \leq t \leq m$ and a $c \in \{-1, 1\}$. We will refer to a vector of this form as a *contiguous vector*. Such perturbations are the central object of study in the problem of differentially private continual counting [13, 10, 4], which provide the following guarantee (proof is in Appendix A).

**Lemma 3.1.** *There exists a randomized algorithm* $\mathsf{CC}_\varepsilon : \mathbb{Z}^m \to \mathbb{Z}^m$ *such that (1) for all vectors* $\mathbf{r}, \tilde{\mathbf{r}} \in \mathbb{Z}^m$ *and* $\mathbf{e} \in \{-1, 0, 1\}^m$ *of the form in Eq.* (1)*, we have*

$$\Pr[\mathsf{CC}_\varepsilon(\mathbf{r}) = \tilde{\mathbf{r}}] \leq e^\varepsilon \Pr[\mathsf{CC}_\varepsilon(\mathbf{r}) = \tilde{\mathbf{r}} + \mathbf{e}], \tag{2}$$

*and (2) for all* $\beta > 0$, $\Pr\left[ \|\mathbf{r} - \mathsf{CC}_\varepsilon(\mathbf{r})\|_\infty \geq 3\log(m)\log(\frac{2m}{\beta})/\varepsilon \right] \leq \beta$.

Consequently, we can set $S_i = x_{(\tilde{r}_i - h)}, \ldots, x_{(\tilde{r}_i + h)}$. Our privacy analysis may then proceed as follows. Let $S'_1, \ldots, S'_m$ denote the slices when $X'$ is used instead of $X$. Now, fix any observed noisy ranks $\tilde{r}_1, \ldots, \tilde{r}_m$ generated from $X$. By the continual counting property (Eq. 2), when using $X'$, it is almost as likely to observe $\tilde{r}_1 + e_1, \ldots, \tilde{r}_m + e_m$, where $e_1, \ldots, e_m$ is the shifting vector such that $S_1, \ldots, S_m$ and $S'_1, \ldots, S'_m$ differ in at most one slice. This setup allows us to analyze the final release $\mathsf{SingleQuantile}(S_1), \ldots, \mathsf{SingleQuantile}(S_m)$ using *parallel composition* — that is, incurring the privacy cost of applying $\mathsf{SingleQuantile}$ only once. Interestingly, the privacy analysis when $X'$ is formed by removing a point from $X$ is more subtle and leads to asymmetric privacy parameters. We discuss this in detail in Section 4.

In summary, the main steps of SliceQuantiles are as follows: first, perturb the target ranks $r_1, \ldots, r_m$ by adding correlated noise generated via continual counting; then, for each slice $S_i = x_{(\tilde{r}_i - h)}, \ldots, x_{(\tilde{r}_i + h)}$, apply the $\mathsf{SingleQuantile}$ mechanism to obtain the output $z_1, \ldots, z_m$.

### 3.2 Implementation Details

Outlined in Algorithm 1, SliceQuantiles is provided with the following parameters: (1) $(w, \varepsilon_1)$, an error bound (to be set later) and privacy budget for CC; (2) $(\ell, \varepsilon_2)$, the minimum list size (to be set later) and privacy budget for SingleQuantile; (3) a probability $\gamma$ of outputting a random value which we assume for now is 0, and (4) data domain bounds $[0, b]$. We can instantiate the algorithm with any SingleQuantile that satisfies $\varepsilon_2$-DP.

A technical challenge arises after sampling the vector $\tilde{\mathbf{r}} = \langle \tilde{r}_1, \ldots, \tilde{r}_m \rangle$: there is no guarantee that the generated slices would be non-overlapping, which would invalidate our privacy analysis [3]. We

---

[3]As a simple counter-example, suppose the $\tilde{r}_i$ all ended up equal. Then, every slice could potentially change for an adjacent dataset, and the privacy parameter could be as high as $m\varepsilon$.

address this as follows. We define the following set for notational convenience:

$$\mathsf{Good}_{m,n,\Delta} = \{(\tilde{r}_1, \ldots, \tilde{r}_m) \in \mathbb{Z}^m : \tilde{r}_1 - \Delta \geq 1, \tilde{r}_i - \tilde{r}_{i-1} > 2\Delta \text{ for } i = 1, \ldots, m-1, \tilde{r}_m \leq n - \Delta\} \ .$$

The set $\mathsf{Good}_{m,n,\Delta}$ consists of noisy vectors with sufficient space around each $\tilde{r}_i$ to ensure that the slices do not intersect. We eliminate the risk of overlap by checking that the sampled noise $\tilde{\mathbf{r}}$ belongs to $\mathsf{Good}_{m,n,h}$ (Line 6), and if not we release a random output from $(0, b)^m$ (corresponding to a failure). To ensure that failure is rare, we require the input ranks $\mathbf{r}$ to belong to $\mathsf{Good}_{m,n,w+h}$ where $w$ is the maximum error introduced by CC (Line 3), which is the precise requirement for the quantile gaps.

A practical improvement proposed in [18] is to adaptively clip the output range of the SingleQuantile algorithm. Our algorithm is able to support this as well as follows. Instead of estimating each $z_i$ using independent calls to SingleQuantile as in Line 12, first compute $z_{m/2}$ for the middle quantile by running SingleQuantile with the entire data domain $[0, b]$. Then, compute $z_{m/4}$ using SingleQuantile with data domain restricted to $[0, z_{m/2}]$, and compute $z_{3m/4}$ using SingleQuantile with data domain restricted to $[z_{m/2}, b]$. Output all quantiles by recursing in a similar binary fashion. The privacy analysis of SliceQuantiles can be modified in a straightforward way to handle these adaptive releases, and in practice, when SingleQuantile is e.g. the exponential mechanism, the utility of SingleQuantile improves by restricting the set of possible outputs.

## 4 Privacy Analysis

We begin by explaining our analysis under approximate differential privacy, and later show how to convert this to a guarantee under pure differential privacy. The proofs for results in this section appear in Appendix B.

**Technical Challenge.** Our first attempt at a privacy proof might proceed as follows: observe that any $\tilde{\mathbf{r}} \in \mathsf{Good}_{m,n,h}$ may be mapped to a corresponding $\tilde{\mathbf{r}}' \in \mathsf{Good}_{m,n,h}$ such that (1) the difference vector $\mathbf{e} = \mathbf{r} - \mathbf{r}'$ is a contiguous vector, and (2) the corresponding slices $S_1, \ldots, S_m$ and $S'_1, \ldots, S'_m$ differ in only one index $j$, and only by a single substitution within that slice. We would then hope to establish $(\varepsilon, \delta)$-differential privacy by arguing the following: (1) a noisy vector sampled by CC belongs to $\mathsf{Good}_{m,n,h}$ with probability at least $1 - \delta$; (2) CC is capable of hiding any binary shift vector $\mathbf{e}$ of the prescribed form; and (3) a single application of $(\varepsilon, \delta)$-differential privacy suffices, since only one slice differs between the adjacent datasets.

Unfortunately, there is a flaw in this argument: if many vectors $\tilde{\mathbf{r}}$ are mapped to the same $\tilde{\mathbf{r}}'$, it becomes difficult to compare the resulting sum over duplicated $\tilde{\mathbf{r}}'$ values to a sum over all $\tilde{\mathbf{r}} \in \mathsf{Good}_{m,n,h}$. A prior work [7] was able to show that there exists a mapping sending at most two distinct values of $\tilde{\mathbf{r}}$ to a single $\tilde{\mathbf{r}}'$. However, this has a significant limitation: even a multiplicative factor of 2 blows up the privacy parameter to $2e^{\varepsilon_1 + \varepsilon_2}$, making it impossible to establish any guarantee with a privacy parameter smaller than $\ln(2)$. (Note that we cannot make use of privacy amplification by subsampling [8, 21] without incurring a large sampling error on quantiles.)

**Key Idea.** A key novelty of our technical proof is to propose a more refined mapping that mitigates the issue above. Specifically, our mapping *injectively* maps each $\tilde{\mathbf{r}} \in \mathsf{Good}_{m,n,h}$ to a corresponding $\tilde{\mathbf{r}}'$ in a slightly larger set. Our mapping depends on whether $X$ is smaller or larger than $X'$. When $X' = X \cup \{x_s\}$ (i.e., an addition), we observe that the mapping above is, in fact, an injection. The difficulty arises only in the case of a removal. Nevertheless, when $X = X' \cup \{x_s\}$, we show that it is possible to construct an injection that ensures at most two slices among $S_1, \ldots, S_m$ and $S'_1, \ldots, S'_m$ differ, each only by a substitution. Formally, we prove the following in Appendix B.1:

**Lemma 4.1.** *Let $X, X'$ denote two adjacent datasets such that $X'$ is smaller than $X$ by exactly one point. Then, there exists a function $F^-(\tilde{\mathbf{r}}) : \mathsf{Good}_{m,n,h} \to \mathsf{Good}_{m,n,h-1}$ such that*

- *$F^-$ is injective.*

- *For $1 \leq i \leq m$, the dataset slices $S_i = x_{(\tilde{r}_i - h)}, \ldots, x_{(\tilde{r}_i + h)}$ and $S'_i = x'_{(\tilde{r}_i - h)}, \ldots, x'_{(\tilde{r}_i + h)}$ satisfy $\sum_{i=1}^m d_{sub}(S_i, S'_i) \leq 2$, where $d_{sub}$ is the number of substitutions of points needed to make $S_i$ and $S'_i$ equal.*

- *$F^-(\tilde{\mathbf{r}}) = \tilde{\mathbf{r}} + \mathbf{e}_{\tilde{\mathbf{r}}}$, where $\mathbf{e}_{\tilde{\mathbf{r}}}$ is binary vector of the form $\mathbf{e}_{\tilde{\mathbf{r}}}[i] = -\mathbf{1}[i \geq j]$ for an index $1 \leq j \leq m+1$.*

*Similarly, if $X'$ is larger than $X$ by exactly one point, then there exists a corresponding function $F^+(\tilde{\mathbf{r}}) : \mathsf{Good}_{m,n,h} \to \mathsf{Good}_{m,n+1,h}$ with the same properties, except that $\mathbf{e}_{\tilde{\mathbf{r}}}$ satisfies $\mathbf{e}_{\tilde{\mathbf{r}}}[i] = \mathbf{1}[i \geq j]$ for an index $1 \leq j \leq m + 1$, and the sum of substitution distances is bounded by 1 instead of 2.*

The fact that $F^+$ and $F^-$ map to slightly different sets than $\mathsf{Good}_{m,n,h}$ is not an important detail; our proof accounts for it by requiring the gap between the input ranks $\mathbf{r}$ to be 1 higher.

As a result, to analyze privacy under add/remove adjacency, we incur the privacy cost of CC once and of SingleQuantile at most twice, yielding a total privacy parameter of $\varepsilon_1 + 2\varepsilon_2$. Due to the asymmetry of our mapping when a point is being added or removed, privacy under substitute adjacency is slightly better since substitution is both an addition and a removal operation. Formally,

**Theorem 4.2.** *Under add/remove adjacency,* SliceQuantiles *with $\gamma = 0$, $w = \frac{3\log(m)\log(2m/\delta)}{\varepsilon_1}$, and any $h \geq 1$, satisfies $(\varepsilon_1 + 2\varepsilon_2, \delta)$-differential privacy. Under substitute adjacency,* SliceQuantiles *satisfies $(2\varepsilon_1 + 3\varepsilon_2, \delta + \delta e^{\varepsilon_1 + 2\varepsilon_2})$-differential privacy.*

The proof is provided in Appendix B.2. Notice that privacy holds for any $h \geq 1$; but our utility results, which we will show later, require $h$ to be sufficiently large.

**Conversion to Pure Differential Privacy.** Observe that $\delta$ plays a limited role in Theorem 4.2, namely it affects only the minimum gap separating the ranks of interest. When the rank gaps (and the data size) are sufficiently large, $\delta$ can be made small enough to be absorbed into the $\varepsilon$ terms. To do this, we prove a reduction from $(\varepsilon, \delta)$-approximate differential privacy to $\varepsilon$-pure differential privacy, which holds when $\delta < \frac{1}{|\mathcal{Y}|}$, the inverse of the size of the output range.

**Lemma 4.3.** *If a mechanism $\mathcal{A}(X)$ with discrete output range $\mathcal{Y}$ satisfies $(\varepsilon, \delta)$-differential privacy with $\frac{\delta|\mathcal{Y}|}{(e^\varepsilon - 1)} \leq 1$, then the mechanism $\tilde{\mathcal{A}}(X)$, which outputs a random sample from $\mathcal{Y}$ with probability $\gamma = \frac{\delta|\mathcal{Y}|}{(e^\varepsilon - 1)}$ and outputs $\mathcal{A}(X)$ otherwise, satisfies $(\varepsilon, 0)$-DP.*

To apply the lemma to SliceQuantiles, we need to discretize the output domain to $[b]^m$ (for example, by rounding to the nearest integers). This introduces a maximum error of 1 in the quantile estimates, which is negligible when the dataset has a minimum gap of 1. A pure differential privacy guarantee is then a corollary of Lemma 4.3 and Theorem 4.2 with $\delta = \frac{\gamma(e^{\varepsilon_1} - 1)}{b^m}$.

**Corollary 4.4.** *Under add/remove adjacency,* SliceQuantiles *with $\gamma > 0$ and*

$$w = \frac{3\log(m)}{\varepsilon_1}\left(m\log b + \log\left(\frac{2m(e^{\varepsilon_2} - 1)}{\gamma}\right)\right)$$

*and with estimates rounded to $\lfloor z_1 \rceil, \ldots, \lfloor z_m \rceil$ satisfies $\varepsilon_1 + 2\varepsilon_2$-pure differential privacy. Under substitute adjacency, Algorithm 1 with $w = \frac{3\log(m)}{\varepsilon_1}\left(m\log b + \log\left(\frac{2m(e^{\varepsilon_1} - 1)}{\gamma}\right) + \varepsilon_1 + 2\varepsilon_2\right)$ satisfies $2\varepsilon_1 + 3\varepsilon_2$-pure differential privacy.*

For most parameter settings, $w$ will dominated by the $m\log b\frac{\log(m)}{\varepsilon_1}$ term. To satisfy the condition $\mathbf{r} \in \mathsf{Good}_{m,n,w+h}$, it is necessary to have $n \geq m^2 \log b \frac{\log(m)}{\varepsilon_2}$. For instance, when $\varepsilon_1 = \varepsilon_2 = 1$, $b = 2^{32}$ and $m = 100$, then $w \approx 65,000$, the minimum gap is $2w = 130,000$, and the total amount of data required is $1.3 \times 10^7$. While the minimum gap between quantiles maybe too large for some datasets, our algorithm offers asymptotic utility improvements over the best-known pure differential privacy algorithms when this assumption is met. We expand on this in the next section.

## 5 Utility Analysis

SliceQuantiles may be implemented with any private algorithm SingleQuantile for estimating a single quantile of a dataset. We introduce a general notion of accuracy for SingleQuantile in order to derive a general error bound.

**Definition 5.1.** *An algorithm* SingleQuantile$(X)$ *is an $(\alpha, \ell, \beta)$ algorithm for median estimation if, for all datasets $X \in [0, b]^n$ of size $n \geq \ell$, with probability at least $1 - \beta$,* SingleQuantile$(X)$ *returns a median estimate $z$ with rank error $|\frac{n}{2} - \mathsf{rank}_X(z)| \leq \alpha$.*

For our purposes, it is sufficient to only require SliceQuantile to return a median estimate, since the median of the slice $S_i$ is the element with the desired rank $\tilde{r}_i$ in $X$. In its general form, our utility guarantee is as follows:

**Theorem 5.1.** *Suppose* SliceQuantiles *is run with an* $(\alpha, \ell, \frac{\beta}{m})$ *algorithm* SingleQuantile *for single quantile estimation. Then, for any input ranks* $r_1, \ldots, r_m$ *such that they are in* $\mathsf{Good}_{n,m,w+h}$ *for* $h = \lceil (\ell - 1)/2 \rceil$, *conditioned on Line 6 not failing, the returned estimates* $Z$ *satisfy* $\mathrm{Err}_X(Q, Z) \leq O\left(\alpha + \frac{\log m \log(\frac{m}{\beta})}{\varepsilon_2}\right)$ *with probability* $1 - \beta$.

We prove this theorem in Appendix C. Next, we specialize this utility theorem in both the pure and approximate DP settings, and compare them to the best-known algorithms.

### 5.1 Utility Guarantee under Pure Differential Privacy

Under pure DP, we may implement SingleQuantile as the exponential mechanism with privacy parameter $\varepsilon_2$ and utility given by the negative rank error. We show in Appendix C that this is a $(h, 2h + 1, \beta)$ algorithm for median estimation with $h = \left\lceil \frac{2 \log(2b/\beta)}{\varepsilon_2} \right\rceil$, for any $\beta \in (0, 1)$. This gives the following immediate corollary (for simplicity, we state it using add/remove adjacency).

**Corollary 5.2.** *Suppose* SliceQuantiles *is run with* $w$ *set as in Corollary 4.4,* SingleQuantile *set to be the exponential mechanism and* $\ell = 2 \left\lceil \frac{2 \log(2bm/\beta)}{\varepsilon_2} \right\rceil + 1$. *Then, the algorithm satisfies* $\varepsilon$-*differential privacy with* $\varepsilon = \varepsilon_1 + 2\varepsilon_2$, *and with probability at least* $1 - \beta - \gamma$, *achieves an error bound of* $\mathrm{Err}_X(Q, Z) \leq O\left(\frac{\log(b)}{\varepsilon} + \frac{\log m \log(m/\beta)}{\varepsilon}\right)$ *for any input quantiles with gap* $\frac{6m \log(b) \log(m)}{\varepsilon_1 n} + O(\frac{\log(m) \log(m(e^\varepsilon - 1)/\gamma)}{\varepsilon n})$.

In contrast, the state-of-the-art algorithm under pure differential privacy attains error $O\left(\frac{\log b \log^2(m)}{\varepsilon} + \frac{\log(m/\beta) \log^2(m)}{\varepsilon}\right)$ [18]. When $\log(b) > \log(m)^2$, error is improved by a factor of $\log(m)^2$. When $\log(m) < \log(b) < \log(m)^2$, the factor is $\log(b)$. This represents an improvement factor of $\min\{\log(b), \log(m)^2\}$ when $b > m$. Note that this improvement comes with a mild constraint: it requires the minimum gap between quantiles to be at least $\Omega(\frac{m \log(b) \log(m)}{\varepsilon n})$.

Nevertheless, the case of equally spaced quantiles remains a well-studied and important problem. In this setting, [18] provide an improved analysis of their algorithm, achieving an error of $O\left(\frac{\log b \log(m)}{\varepsilon} + \frac{\log(m/\beta) \log(m)}{\varepsilon}\right)$. Our algorithm still offer an improvement by a factor of $\log(m)$ for the case $b > m$. Note that to meet the required quantile gap, it must hold that $n \geq \Omega(m^2 \log b \frac{\log m}{\varepsilon})$.

### 5.2 Finding Quantiles Under Approximate Differential Privacy

Under approximate differential privacy, quantile estimation algorithms with more favourable dependence on $b$ are known. Specifically, as shown in [11], there exists an $(\varepsilon, \delta)$-DP algorithm which is a $(\frac{\ell}{2}, \ell, \delta \log^*(b))$ algorithm for median estimation with $\ell = \frac{1000 \log^*(b)}{\varepsilon} \log(\frac{1}{\delta})^2$. This algorithm may be used to answer general *threshold queries*, or the fraction of data below a query point $x \in [0, b]$, of a dataset of size $n$, and provides error that scales with $\log^*(b)$ instead of $\log(b)$. We provide the full details of these results in Appendix C. When translated to quantile estimation, this threshold query-based method can answer any set of quantiles with error $O(\log^*(b) \frac{\log^2(\log^*(b)/\beta\delta)}{\varepsilon})$. We note that there is no dependence on $m$ in this bound.

Though the factor of 1000 above is probably far from tight, even the $\log^2(1/\delta)$ factor incurred by the algorithm can be quite large, and result in higher error despite the improved $\log^*(b)$ dependence on the domain size. To avoid incurring $\log(1/\delta)$ terms, we instantiate the $(\varepsilon, \delta)$ version of SliceQuantiles with the exponential mechanism to obtain:

**Corollary 5.3.** *Suppose* SliceQuantiles *is run with* $w$ *set as in Theorem 4.2,* SingleQuantile *set to be the exponential mechanism, with* $\gamma = 0$ *and* $\ell = 2 \left\lceil \frac{2 \log(2bm/\beta)}{\varepsilon_2} \right\rceil + 1$. *Then, the algorithm satisfies* $(\varepsilon, \delta)$-*differential privacy with* $\varepsilon = \varepsilon_1 + 2\varepsilon_2$, *and with probability at least* $1 - \beta - \delta$,

*achieves an error bound of* $\mathrm{Err}_X(Q, Z) \leq O(\frac{\log(b)}{\varepsilon} + \frac{\log(m)\log(\frac{m}{\beta})}{\varepsilon})$ *for any input quantiles with gap* $\frac{6\log(m)\log(2m/\delta)}{\varepsilon_1 n} + \frac{4\log(2bm/\beta)}{\varepsilon_2 n}$.

Observe that whenever $\log m \log(\frac{1}{\delta}) \leq \log b$, the gap is actually asymptotically *less* than the normalized error bound. This means the gap requirement can be removed entirely by merging any quantiles too close, and this will only increase the error term by a constant factor.

Compared with the previous error guarantee, the guarantee in Corollary 5.3 is in fact lower whenever $\log(b) < \log^2(\frac{1}{\delta})\log^*(b)$ and $\log(m)$ is sufficiently small compared to $\log(1/\delta)$, which is often true for practical choices of the parameters. For instance, $\log^*(b)$ rarely exceeds 4 even for very large domains, while $\log(b)$ is typically below 64, and $\log^2(1/\delta)$ often reaches into the thousands for typical choices of $\delta$. Furthermore, the hidden constant factor in the former algorithm is not known to be under 1000, while in our case it is roughly 10. These factors underscore the superior practical performance of our approach.

## 6 Experiments

For our experiments (code is open-sourced[4]), we use a variant of the $k$-ary tree CC mechanism introduced in [4] with two-sided geometric noise, and SingleQuantile implemented as the exponential mechanism [18, 20]. Similarly to Kaplan et al. [18], we construct two real-valued datasets by adding small Gaussian noise to the AdultAge and AdultHours datasets [5]; both datasets, corresponding to ages and hours worked per week, exist on the interval $[0, 100]$; we use this as our data domain. Unlike their approach, however, we scale the dataset by duplicating each data point 12 times (preserving multiset ranks), resulting in approximately $n = 500{,}000$ entries. This allows us to analyze a large number of quantiles without needing to resort to merging techniques. Concretely, the gap assumption for our parameters is 0.005, so up to 200 equally-spaced quantiles could be answered. The distribution of these datasets are detailed in Appendix E. To ensure a minimum spacing of $1/n$ between data points, we add $i/n$ to the $i$th element in the sorted dataset. For each $m \in \{10, 20, \dots, 200\}$, we randomly sample $m$ quantiles from the set of 250 uniformly spaced quantiles $\{i/251 : i = 1, \dots, 250\}$, and run experiments on both datasets. This sampling procedure is performed independently for each experiment, ensuring that the reported results represent an average over both good and bad instantiations of the problem. We evaluate the mechanism under both substitute and add/remove adjacency, using $\varepsilon = 1$ and $\delta = 10^{-16} \ll \frac{1}{n^2}$ as the privacy parameters. Additional experiments with varying privacy budgets are presented in Appendix E. The $y$-axis in our results reports the average maximum rank error, with 95% confidence intervals computed via bootstrapping over 200 experiments. Results are shown in Figure 1: the first two plots (from the left) correspond to substitute adjacency, while the remaining plots correspond to add/remove adjacency.

**Baseline Algorithms.** Our primary baseline is Approximate Quantiles [18], which we abbreviate to AQ. Experiments in that reference demonstrate improved utility over the AppindExp algorithm [15], hence we do not include AppindExp in our comparisons. AQ satisfies both pure and approximate DP, with the approximate DP analysis leveraging an improved analysis of the exponential mechanism under zero-concentrated DP [6, 9]. As we are using approximate DP for our experiments, we include privacy analyses of AQ as comparison baselines. We also compare to the histogram-based method of [19]—because its error is much higher due to making linear approximations of the data distribution, we put these plots in Appendix E.

**Implementation details of** SliceQuantiles**.** Our empirical results indicate that the most effective strategy for allocating the privacy budget between CC and SingleQuantile is to divide it equally, assigning half to each mechanism. To compute the size of the slice, we use $h = \left\lceil \frac{2}{\varepsilon} \log\left(\frac{2m\psi}{\beta}\right) \right\rceil$, according to Theorem C.1, with $\beta = 0.05$ and $\psi = \frac{b-a}{g} = 100n$ as $[a, b] = [0, 100]$ and $g = \frac{1}{n}$. We used numerical optimization of the Chernoff bound to compute the smallest possible value of the parameter $w$ bounding the CC mechanism with failure $\delta$, beyond the asymptotic expression given in Lemma 3.1. The details of this optimization appear in Appendix D. This reduced the quantile gap assumption and maximized the number of quantiles that SliceQuantiles is able to answer.

---

[4]https://github.com/NynsenFaber/DP_CC_quantiles

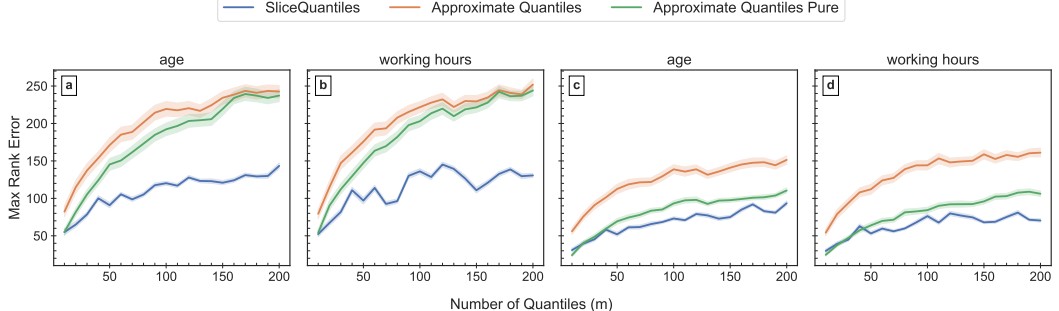

Figure 1: Experiments on `AdultAge` and `AdultHours` datasets. The datasets contain approximately $5 \cdot 10^5$ data points. Each experiment was run 200 times, with each run using a random sample from a set of 250 uniformly spaced quantiles. Plots $a$ and $b$ are for substitute adjacency, while $c$ and $d$ correspond to add/remove adjacency. Approximate Quantiles in the above figure refers to the algorithm (AQ) from [18]. The privacy settings are: $(1, 10^{-16})$-DP for SliceQuantiles and AQ, and $(1, 0)$-DP for AQ with pure DP guarantee [18].

**Results.** We plot the three algorithms in Figure 1. The plots indicate that SliceQuantiles performs better than AQ, with a notable advantage under substitute adjacency. This is in line with our theoretical argument on a tighter bound under this adjacency (Theorem 4.2). Consistent with our prior observations, AQ under approximate differential privacy performs worse than AQ with pure differential privacy, due to our lower choice of $\delta$. Because its utility guarantee is independent of $\delta$, SliceQuantiles is able to circumvent this issue.

## 7 Conclusion

In this paper, we have proposed new mechanisms for approximating multiple quantiles on a dataset, satisfying both $\varepsilon$ and $(\varepsilon, \delta)$ differential privacy. As long as the minimum gap between the queried quantiles is sufficiently large, the mechanisms achieve error with a better dependence on the number of quantiles and $\delta$ than prior work. Our experimental results demonstrate that these mechanisms outperform prior work in practice, in particular when the number of quantiles is large. An interesting question for future directions is to explore if a more careful analysis could reduce the minimum gap requirement or if other practical mechanisms for differentially private quantiles could further improve accuracy of computing many quantiles privately in practice.

## Acknowledgments

Anders Aamand and Rasmus Pagh were supported by the VILLUM Foundation grant 54451.

Jacob Imola and Rasmus Pagh were supported by a Data Science Distinguished Investigator grant from Novo Nordisk Fonden

Fabrizio Boninsegna was supported in part by the MUR PRIN 20174LF3T8 AHeAD project, and by MUR PNRR CN00000013 National Center for HPC, Big Data and Quantum Computing.

The research described in this paper has also received funding from the European Research Council (ERC) under the European Union's Horizon 2020 research and innovation programme under grant agreement No 803096 (SPEC) and the Danish Independent Research Council under Grant-ID DFF-2064-00016B (YOSO).

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

# A  Details From Continual Counting

In summary, the binary tree mechanism [13, 10] achieves $\varepsilon$-DP when the input vector $\mathbf{r}$ is changed to $\mathbf{r} + \mathbf{e}$, where $\mathbf{e}$ is a contiguous $0/1$ or $0/-1$-valued vector. The mechanism achieves this by constructing a segment tree whose leaves are the intervals $[0, 1), \ldots, [m-1, m)$, and sampling a Laplace random variable $\eta_I$ for each node $I$ of the tree. For $T = \lceil \log_2(m+1) \rceil$, let $I_1, \ldots, I_T$ denote the interval decomposition of $[0, i)$. Then, the estimate for $\tilde{r}_i$ is given by $r_i + \sum_{j=1}^{T} \eta_{I_j}$ rounded to the nearest integer (by convention, for integers $k$ the number $k - 1/2$ is rounded up to $k$). The correlated noise being added to each $r_i$ allows the total error to grow only logarithmically with $m$. The final rounding ensures that the noisy ranks are integer valued. The rounding is private due to post processing and it only incurs an additional rank error of at most $1/2$

## A.1  Proof of Lemma 3.1

We start with the following concentration lemma:

**Corollary A.1** (From Corollary 2.9 of [10])**.** *Suppose $\gamma_i$'s are independent random variables, where each $\gamma_i$ has Laplace distribution $\mathsf{Lap}(b_i)$. Suppose $Y = \sum_i \gamma_i$, $b_M = \max_i b_i$ and $\delta \in (0, 1)$. Let $\nu = \max\{\sqrt{\sum_i b_i^2}, b_M \sqrt{\ln(2/\delta)}\}$. Then $\Pr[|Y| > \nu \sqrt{8 \ln(2/\delta)}] \leq \delta$.*

*Proof of Lemma 3.1.* Define $\mathtt{round} : \mathbb{R}^m \to \mathbb{Z}^m$ to be the rounding function rounding each coordinate of a vector $s \in \mathbb{R}^m$ to the nearest integer (rounding up in case of ties). Note that $\mathtt{round}(s + x) = \mathtt{round}(s) + x$ whenever $x \in \mathbb{Z}^m$. To prove the first guarantee, observe that since the binary tree mechanism is an additive noise mechanism (i.e. $\mathsf{CC}(\mathbf{r}) = \mathtt{round}(\mathbf{r} + \mathcal{N})$, where $\mathcal{N}$ is a noisy vector) the privacy guarantee of the binary tree mechanism implies that

$$\Pr[\mathsf{CC}(\mathbf{r}) = \tilde{\mathbf{r}}] = \Pr[\mathtt{round}(\mathbf{r} + \mathcal{N}) = \tilde{\mathbf{r}}] \leq e^\varepsilon \Pr[\mathtt{round}(\mathbf{r} - \mathbf{e} + \mathcal{N}) = \tilde{\mathbf{r}}] = e^\varepsilon \Pr[\mathsf{CC}(\mathbf{r}) = \tilde{\mathbf{r}} + \mathbf{e}]$$

To show utility, each coordinate of $\mathcal{N}$ is the sum of at most $T$ independent Laplace random variables with variance $T^2/\varepsilon^2$ to each estimate. From Corollary A.1 we have that, with probability at least $1 - \beta/m$, each estimate has an error at most $\frac{T}{\varepsilon} \sqrt{8 \ln(2m/\beta)} \max\{\sqrt{T}, \sqrt{\ln(2m/\beta)}\}$. The claim follows by a union bound and analysing the asymptotic of $\frac{T}{\varepsilon} \sqrt{8 \ln(2m/\beta)} \max\{\sqrt{T}, \sqrt{\ln(2m/\beta)}\} \leq \frac{T}{\varepsilon} \sqrt{8 \ln(2m/\beta)}(\sqrt{T} + \sqrt{\ln(2m/\beta)})$. $\qquad \square$

# B  Omitted Proofs From Section 4

## B.1  Proof of Lemma 4.1

*Proof.* Suppose first that $X'$ adds a point to $X$. This means that there is a minimal index $s$ such that $x_{(i)} = x'_{(i)}$ for all $i < s$, and $x'_{(i+1)} = x_{(i)}$ for all $i \geq s$. We will define $F^+(\tilde{\mathbf{r}}) = \tilde{\mathbf{r}} + \mathbf{e}_{\tilde{\mathbf{r}}}$, where each coordinate of $\mathbf{e}_{\tilde{\mathbf{r}}}$ is defined by

$$e_i = \begin{cases} 0 & \tilde{r}_i - h \leq s \\ 1 & \tilde{r}_i - h > s. \end{cases}$$

It is clear that this vector belongs to $\mathsf{Good}_{m, n+1, h}$ and that Property (3) of the map is satisfied. To see injectivity, observe that if $\tilde{\mathbf{r}}, \tilde{\mathbf{r}}'$ are different, but mapped to the same output, then they must have different vectors $\mathbf{e}, \mathbf{e}'$. Furthermore, the coordinates of $\tilde{\mathbf{r}}, \tilde{\mathbf{r}}'$ only differ by 1. These two things can only happen if $\tilde{r}_{i^*} - h = s$ and $\tilde{r}'_{i^*} - h = s + 1$ for an index $i^*$. However, this will then produce $e_{i^*} = 0$ and $e'_{i^*} = 1$, resulting in the $i^*$ coordinates of $F^+(\tilde{\mathbf{r}}), F^+(\tilde{\mathbf{r}}')$ to be $h + s$ and $h + s + 2$, respectively, making it impossible for equality.

Finally, Property (2) holds because the slices $S_i, S'_i$ will disagree only if $\tilde{r}_i \in [s - h, s + h]$, and the sum of substitution distances is 1.

In the case that $X'$ removes a point from $X$, then $x_{(i)} = x'_{(i)}$ for all $i < s$ and $x_{(i)} = x'_{(i-1)}$ for all $i \geq s$. We define the map $F^-(\tilde{\mathbf{r}}) = \tilde{\mathbf{r}} + \mathbf{e}_{\tilde{\mathbf{r}}}$, where each coordinate of $\mathbf{e}_{\tilde{\mathbf{r}}}$ is instead defined by

$$e_i = \begin{cases} 0 & i = 1 \vee \tilde{r}_{i-1} - h \leq s \\ -1 & i > 1 \wedge \tilde{r}_{i-1} - h > s. \end{cases}$$

Again, Property (3) and membership in $\mathsf{Good}_{m,n,h-1}$ is immediate. To argue injectivity, observe that if $\tilde{\mathbf{r}}, \tilde{\mathbf{r}}'$ are different, but mapped to the same output, then they must have different vectors $\mathbf{e}, \mathbf{e}'$. This is only possible if $\tilde{r}_{i^*-1} - h = s$ and $\tilde{r}'_{i^*-1} - h = s+1$ for some index $i^* > 1$. However, this then implies that $e_{i^*-1} = e'_{i^*-1} = 0$, resulting in $F^-(\tilde{r}_{i^*}) = \tilde{r}_{i^*}$ and $F^-(\tilde{r}'_{i^*-1}) = \tilde{r}'_{i^*-1}$ and forcing the maps to still have different outputs $F^-(\tilde{r}_{i^*-1}) \neq F^-(\tilde{r}'_{i^*})$.

Property (2) follows because the slices $S_i, S'_i$ disagree only in potentially two locations; namely the index $i^*$ where $a \in [\tilde{r}_{i^*} - h, \tilde{r}_{i^*} + h]$ (if it exists), and the index $j^*$ which is the minimum index such that $a < \tilde{r}_{j^*} - h$. Each disagreement adds one to the substitution distance, and thus the total sum is at most 2.

Note that at first glance, it may appear as though the case where $X'$ removes a point from $X$ could be solved analogously to the case where $X'$ adds a point, defining $e_i$ as:

$$e_i = \begin{cases} 0 & \tilde{r}_i - h \leq s \\ -1 & \tilde{r}_i - h > s. \end{cases}$$

However, this choice would not lead to an injective mapping. In particular, we can set $\tilde{r}_{i^*} - h = s$ and $\tilde{r}'_{i^*} - h = s+1$ for an index $i^*$, which are different, but mapped to the same output. Specifically, this will produce $e_{i^*} = 0$ and $e'_{i^*} = -1$, resulting in $F^-(\tilde{r}_{i^*}) = s + h = F^-(\tilde{r}'_{i^*})$.

$\square$

## B.2  Proof of Theorem 4.2

*Proof.* We will first prove add/remove privacy.

Let $\mathcal{A}(X)$ denote $\mathsf{SliceQuantiles}$ run on input $X$, and let $\mathcal{A}(X, \tilde{\mathbf{r}})$ denote the algorithm conditioned on the noisy ranks $\tilde{\mathbf{r}} = (\tilde{r}_1, \ldots, \tilde{r}_m)$. Let $F^+, F^-$ denote the maps guaranteed by Lemma 4.1. We will first assume that $X'$ is larger than $X$, and thus we will use $F^+$ in the following. For any output set $Z$, we have

$$
\begin{aligned}
\Pr[\mathcal{A}(X) \in Z] &= \sum_{\tilde{\mathbf{r}} \in \mathbb{Z}^m} \Pr\left[\mathcal{A}(X, \tilde{\mathbf{r}}) \in Z\right] \Pr[\mathsf{CC}(\mathbf{r}) = \tilde{\mathbf{r}}] \\
&\leq \Pr[\mathsf{CC}(\mathbf{r}) \notin \mathsf{Good}_{m,n,h}] + \sum_{\tilde{\mathbf{r}} \in \mathsf{Good}_{m,n,h}} \Pr\left[\mathcal{A}(X, \tilde{\mathbf{r}}) \in Z\right] \Pr[\mathsf{CC}(\mathbf{r}) = \tilde{\mathbf{r}}] \\
&\leq \delta + \sum_{\tilde{\mathbf{r}} \in \mathsf{Good}_{m,n,h}} \Pr\left[\mathcal{A}(X, \tilde{\mathbf{r}}) \in Z\right] \Pr[\mathsf{CC}(\mathbf{r}) = \tilde{\mathbf{r}}] \\
&\leq \delta + \sum_{\tilde{\mathbf{r}} \in \mathsf{Good}_{m,n,h}} e^{\varepsilon_2} \Pr\left[\mathcal{A}(X, F^+(\tilde{\mathbf{r}})) \in Z\right] e^{\varepsilon_1} \Pr[\mathsf{CC}(\mathbf{r}) = F^+(\tilde{\mathbf{r}})] \\
&\leq \delta + e^{\varepsilon_1 + \varepsilon_2} \sum_{\tilde{\mathbf{r}} \in \mathsf{Good}_{m,n+1,h}} \Pr\left[\mathcal{A}(X', \tilde{\mathbf{r}}) \in Z\right] \Pr[\mathsf{CC}(\mathbf{r}) = \tilde{\mathbf{r}}] \\
&\leq \delta + e^{\varepsilon_1 + \varepsilon_2} \Pr[\mathcal{A}(X') \in Z],
\end{aligned}
$$

where the third line follows from Lemma 3.1, which shows $\Pr[\|\mathbf{r} - \tilde{\mathbf{r}}\|_\infty \geq w] \leq \delta$, and by the assumption that $\mathbf{r} \in \mathsf{Good}_{m,n,h+w+1}$; the fourth follows from Properties (2) and (3) of Lemma 4.1 along with Lemma 3.1; and the fifth follows from Property (1) of Lemma 4.1. When $X'$ is smaller

than $X$, then we use the map $F^-$, and we obtain

$$\Pr[\mathcal{A}(X) \in Z] = \sum_{\tilde{\mathbf{r}} \in \mathbb{Z}^m} \Pr[\mathcal{A}(X, \tilde{\mathbf{r}}) \in Z] \Pr[\mathsf{CC}(\mathbf{r}) = \tilde{\mathbf{r}}]$$

$$\leq \Pr[\mathsf{CC}(\mathbf{r}) \notin \mathsf{Good}_{m,n-1,h+1}] + \sum_{\tilde{\mathbf{r}} \in \mathsf{Good}_{m,n-1,h+1}} \Pr[\mathcal{A}(X, \tilde{\mathbf{r}}) \in Z] \Pr[\mathsf{CC}(\mathbf{r}) = \tilde{\mathbf{r}}]$$

$$\leq \delta + \sum_{\tilde{\mathbf{r}} \in \mathsf{Good}_{m,n-1,h+1}} \Pr[\mathcal{A}(X, \tilde{\mathbf{r}}) \in Z] \Pr[\mathsf{CC}(\mathbf{r}) = \tilde{\mathbf{r}}]$$

$$\leq \delta + \sum_{\tilde{\mathbf{r}} \in \mathsf{Good}_{m,n-1,h+1}} e^{2\varepsilon_2} \Pr[\mathcal{A}(X, F^-(\tilde{\mathbf{r}})) \in Z] e^{\varepsilon_1} \Pr[\mathsf{CC}(\mathbf{r}) = F^-(\tilde{\mathbf{r}})]$$

$$\leq \delta + e^{\varepsilon_1 + 2\varepsilon_2} \sum_{\tilde{\mathbf{r}} \in \mathsf{Good}_{m,n-1,h}} \Pr[\mathcal{A}(X', \tilde{\mathbf{r}}) \in Z] \Pr[\mathsf{CC}(\mathbf{r}) = \tilde{\mathbf{r}}]$$

$$\leq \delta + e^{\varepsilon_1 + 2\varepsilon_2} \Pr[\mathcal{A}(X') \in Z],$$

where the deductions are the same, and the only change is the final parameter is $\varepsilon_1 + 2\varepsilon_2$ as the constant is higher.

To prove substitution privacy, we may simply use the fact that for two neighboring datasets $X, X'$, there is a dataset $X_1$ such that $X_1$ may be obtained from either $X, X'$ by removing a point. Thus, from what we've already shown, the pair $X, X_1$ satisfies $(\varepsilon_1 + 2\varepsilon_2, \delta)$-DP, while $X_1, X'$ satisfies $(\varepsilon_1 + \varepsilon_2, \delta)$-DP. By group privacy, we have a final guarantee of $(2\varepsilon_1 + 3\varepsilon_2, \delta + \delta e^{\varepsilon_1 + 2\varepsilon_2})$  □

### B.3   Proof of Lemma 4.3

*Proof.* By the $(\varepsilon, \delta)$-DP guarantee, the probability of observing any $y \in \mathcal{Y}$ may be bounded by

$$\Pr[\mathcal{A}(X) = y] \leq e^\varepsilon \Pr[\mathcal{A}(X') = y] + \delta.$$

Now, we have

$$\Pr[\tilde{\mathcal{A}}(X) = y] = (1 - \gamma) \Pr[\mathcal{A}(X) = y] + \gamma \frac{1}{|\mathcal{Y}|}$$

$$\leq (1 - \gamma)(e^\varepsilon \Pr[\mathcal{A}(X') = y] + \delta) + \gamma \frac{1}{|\mathcal{Y}|}$$

$$= e^\varepsilon \left( (1 - \gamma)(\Pr[\mathcal{A}(X') = y]) + \gamma \frac{1}{|\mathcal{Y}|} \right) + \delta(1 - \gamma) + \gamma(1 - e^\varepsilon) \frac{1}{|\mathcal{Y}|}$$

$$= e^\varepsilon \Pr[\tilde{\mathcal{A}}(X') = y] + \delta(1 - \gamma) + \gamma(1 - e^\varepsilon) \frac{1}{|\mathcal{Y}|}$$

$$\leq e^\varepsilon \Pr[\tilde{\mathcal{A}}(X') = y] + \delta + \gamma(1 - e^\varepsilon) \frac{1}{|\mathcal{Y}|}$$

$$= e^\varepsilon \Pr[\tilde{\mathcal{A}}(X') = y].$$

□

## C   Omitted Details From Section 5

### C.1   Proof of Theorem 5.1

*Proof.* By a union bound, conditioned on Line 6 not failing, with probability at least $1 - \beta$, the returned quantiles $z_1, \ldots, z_m$ will satisfy $|\mathsf{rank}_X(z_i) - \tilde{r}_i| \leq \alpha$. By Lemma 3.1, each $\tilde{r}_i$ satisfies $\|\mathbf{r} - \tilde{\mathbf{r}}\|_\infty \leq \frac{3 \log(m)}{\varepsilon} \log(\frac{m}{2\beta})$ with probability at least $1 - \beta$. The bound follows from the triangle inequality.  □

### C.2   Details on Exponential Mechanism

Assuming that each point lies in $[a, b]$, the exponential mechanism samples a point $z \in [a, b]$ with probability $\exp(-\frac{\varepsilon}{2}\mathsf{Err}_X(q, Z))$—here we will assume the quantile $q = \frac{1}{2}$ for the median. This can

be implemented by sampling an interval $I_k = [x_{(k)}, x_{(k+1)}]$, with $x_{(0)} = a$ and $x_{(k+1)} = b$, with probability proportional to $\exp\left(-\frac{\varepsilon}{2}|rn - k|\right)|I_k|$, and then releasing a value uniformly sampled from the selected interval (see Appendix A in [18]).

The utility of the exponential mechanism for median estimation is as follows:

**Theorem C.1.** *Given a dataset $X \in [a, b]^n$ with minimum gap $g > 0$, parameters $\beta \in (0, 1)$ and $\varepsilon > 0$, let $\psi = \frac{b-a}{g}$. Then, the exponential mechanism is a $(h, 2h, \beta)$ algorithm for median estimation for $h = \left\lceil \frac{2}{\varepsilon} \log\left(\frac{2\psi}{\beta}\right) \right\rceil$.*

*Proof.* It is sufficient to suppose the dataset has size $2h$, and to bound the probability of sampling in the interval $[a, x_{(1)}]$ and $[x_{(2h)}, b]$. Let $\mathcal{I} = \{[a, x_{(1)}], [x_{(1)}, x_{(2)}], \ldots, [x_{(2h)}, b]\}$ be the set of intervals sampled by the exponential mechanism. The probability the sample $z$ lies in the interval $[a, x_{(r-h)}]$ is

$$\Pr[z \in [0, x_{(1)}]] = \frac{e^{-\frac{\varepsilon}{2}h}(x_{(1)} - a)}{\sum_{i=1}^{2h} e^{-\frac{\varepsilon}{2}|i-h|}(x_{(i)} - x_{(i-1)})} \leq \frac{b-a}{g}e^{-\frac{\varepsilon}{2}h}$$

as $\sum_{i=1}^{2h} e^{-\frac{\varepsilon}{2}|i-h|}(x_{(i)} - x_{(i-1)}) \geq (x_{(h+1)} - x_{(h)}) \geq g$. By our choice of $h$, this probability is at most $\frac{\beta}{2}$. The same upper bound can be found for the other extreme interval $[x_{(2h)}, b]$, and the result follows from a union bound. $\square$

### C.3 Details on Threshold Queries

We discuss how the existing work on threshold queries may be used to answer quantiles. These algorithms actually work by solving a much simpler interior point problem, where the goal on an input dataset $X$, is to simply return a point $z$ such that $x_{(1)} \leq z \leq x_{(n)}$. For a small enough dataset, the interior point algorithm can also provide a median estimate with low error.

**Lemma C.2.** *(Adapted from Theorem 3.7 of [11]): There exists an $(\varepsilon, \delta)$-DP algorithm $\mathcal{A}_{int}$ which is a $\left(\frac{500 \log^* |X|}{\varepsilon} \log(\frac{1}{\delta})^2, \frac{1000 \log^* |X|}{\varepsilon} \log(\frac{1}{\delta})^2, \delta \log^* |X|\right)$ algorithm for median estimation.*

This demonstrates that it is possible to instantiate SingleQuantile with a better dependence on $b$, though the best-known constant factor of 1000 means it is not a practical improvement.

The interior point algorithm may be used to answer threshold queries, which are queries of the form $F_X(z) = \frac{1}{n}\text{rank}_X(z)$, with the following guarantee:

**Lemma C.3.** *(Adaptive from Theorem 3.9 of [11]): There exists an $(\varepsilon, \delta)$-DP algorithm which, with probability $1 - \beta$, can answer any threshold queries $F_X(z)$ with error $O(\frac{\log^*(b) \log(\frac{\log^*(b)}{\beta\delta})^2}{\varepsilon n})$.*

Using binary search, this algorithm may be used to answer any set of $Q$ quantiles to within error $O(\frac{\log^*(b) \log(\frac{\log^*(b)}{\beta\delta})^2}{\varepsilon n})$.

## D Better Computation of the Maximum Error for Continual Counting

In this section, we describe the procedure used to compute the maximum absolute error of the continual counting mechanism used in our experiments. The analysis relies on applying a Chernoff bound to the mechanism and subsequently determining numerically the value that minimizes the error.

The mechanism under consideration is a variant of the approach developed in [4], which makes use of a $k$-ary tree structure to introduce correlated noise. Our modification lies in sampling the noise from a discrete Laplace distribution. Let $k > 0$, the work in [4] provides guidelines for choosing $k$ in order to minimize the worst-case variance, and let $T = \lceil \log_k(m + 1) \rceil$ denote the depth of the tree. The noise at each node of the tree is $\eta_i \sim \text{DL}(b)$ where $b = e^{-\varepsilon/T}$. The probability mass function of the discrete Laplace distribution is given by

$$\Pr_{y \sim \text{DL}(b)}[y = x] = \frac{1-b}{1+b}b^{|x|}.$$

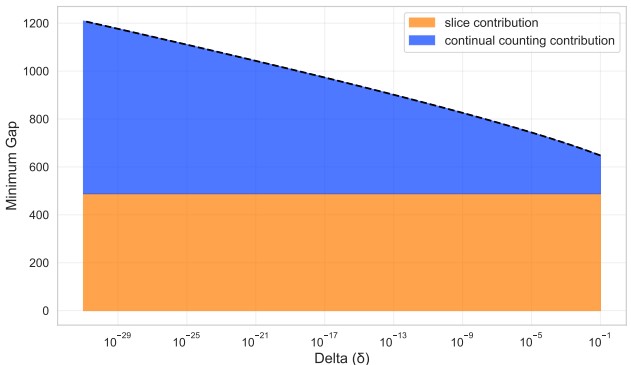

Figure 2: The plot illustrates the minimum gap on the input ranks that is required to achieve $(1, \delta)$-differential privacy. Two distinct contributions are visible: the slice contribution and the continual counting contribution. The slice contribution corresponds to the minimum slice size required to ensure that all `SingleQuantile` instances succeed with probability at least $0.95$. Importantly, this term is independent of $\delta$. In contrast, the continual counting contribution grows as $\delta$ decreases. The computation considers 100 quantiles and add/remove privacy.

Each continual counting noise $Z_i$, for $i \in [m]$, is the sum of at most $T$ discrete Laplace noises. Using a Chernoff bound, we get for any $\lambda > 0$

$$\Pr\left[|Z_i| \geq \mathcal{E}\right] \leq 2e^{-\lambda \mathcal{E} + T \log(M_\eta(\lambda))}$$

where $M_\eta(\lambda)$ is the moment generating function of $\eta$ which is sampled from $\text{DL}(b)$ (see [16])

$$M_\eta(\lambda) = \frac{(1-b)^2}{(1-e^{-\lambda}b)(1-e^{\lambda}b)}.$$

By using a union bound over $m$ continual counting noises we obtain

$$\Pr\left[\max_{i \in [m]} |Z_i| \geq \mathcal{E}\right] \leq 2m e^{-\lambda \mathcal{E} + T \log(M_\eta(\lambda))} = \delta.$$

Given $\delta > 0$, finding $\lambda \in (0, -\log b)$ (so that the moment generating function is positive) such that $\mathcal{E}$ is minimum cannot be solved analytically. Our linear search uses 100 different $\lambda$ in $[10^{-6}, -\log(b) \cdot 0.99]$ with an equal space and for each computes $\mathcal{E}(\lambda)$

$$\mathcal{E}(\lambda) = \frac{\log(2m/\delta) + T \log(M_\eta(\lambda))}{\lambda},$$

the minimum error is then released.

### D.1 Relation Between $\delta$ and Minimum Gap

To apply `SliceQuantiles` it is necessary that the input ranks are $r_1, \ldots, r_m \in \text{Good}_{m,n,w+h}$. Thus, the minimum gap between ranks must be $\min_{i \neq j} |r_i - r_j| > 2(w + h)$. While $h$ can be computed using Theorem C.1, $w$ is computed following the procedure illustrated in the previous section. Figure 2 shows the minimum gap required, $2(w + h)$, for different values of $\delta$. The results consider the case of add/remove privacy, $\varepsilon = 1$ and $m = 100$ (number of quantiles).

## E Additional Experimental Material

In this section we give further experimental results. In Figure 3 the density and the cumulative distribution of the two datasets are depicted. The distributions are shown after data pre-processing, which accounts for data augmentation to increase the minimum gap among ranks, the insertion of low variance Gaussian noise to ensure uniqueness of the data points, and translation, thus the addition of $i/n$ (where $n$ is the size of the augmented dataset) to each point $x_{(i)}$ so to guarantee that $\min_{i \neq j} |x_i - x_j| \geq 1/n$. This last features allows to set $g = 1/n$ when computing the slicing parameter using Theorem C.1.

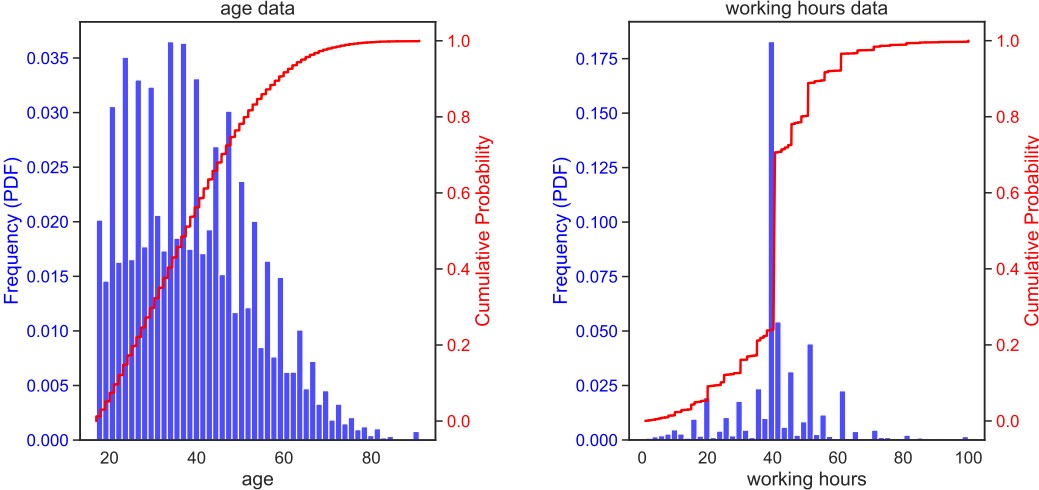

Figure 3: Histogram representation and cumulative distribution of AdultAge and `AdultHours` after pre-processing (data augmentation, Gaussian noise addition, and translation).

As a new baseline, we include the histogram density estimator algorithm, denoted as Hist, introduced in [19]. This algorithm employs a differentially private estimate of the cumulative distribution function, obtained by injecting Laplace noise into a histogram representation of the dataset, to compute quantiles. Although the algorithm is conceptually simple, it requires the bin size to be determined in advance, which directly influences the utility of the resulting estimates. Given that the dataset bounds are known, achieving uniform bin sizes reduces to selecting the number of bins. In these experiments, we consider three configurations for the number of bins: $\frac{N}{10}$, $\frac{N}{2}$, and $N$.

We run the same experiments with additional privacy budget $\varepsilon = 0.5$ and $\varepsilon = 5$, to study the behavior in the small and high privacy regime. Figure 4 depicts these experiments, showing that, if the data set is sufficiently large, SliceQuantiles achieves smaller error than AQ from [18]. In contrast, the performance of Hist varies depending on the chosen number of bins, yet it consistently exhibits an error that is approximately one order of magnitude higher.

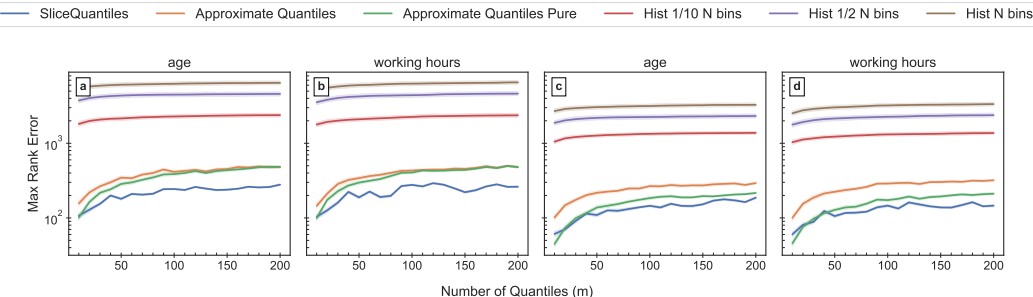

(a) Experiments with $(0.5, 10^{-16})$-DP for SliceQuantiles and AQ, while $(0.5, 0)$-DP for AQ with pure DP accounting and Hist. Such small privacy budget requires a large minimum gap between ranks, thus, we augmented the dataset $24$ times obtaining $1172208$ data points. Plots $a$ and $b$ are for substitute adjacency, while $c$ and $d$ correspond to add/remove adjacency.

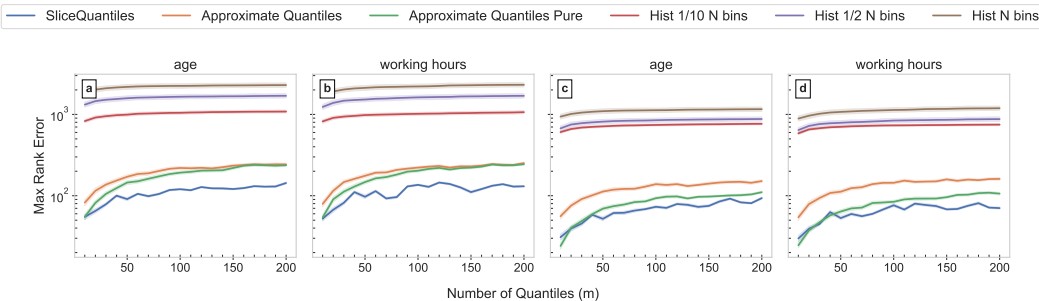

(b) Experiments with $(1, 10^{-16})$-DP for SliceQuantiles and AQ, while $(1, 0)$-DP for AQ with pure DP accounting and Hist. For these privacy budget we have to increase the dataset $12$ times obtaining $586104$ data points. Plots $a$ and $b$ are for substitute adjacency, while $c$ and $d$ correspond to add/remove adjacency.

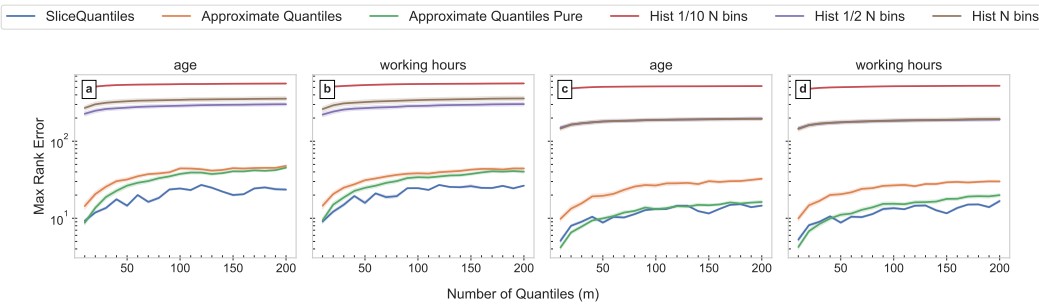

(c) Experiments with $(5, 10^{-16})$-DP for SliceQuantiles and AQ, while $(5, 0)$-DP for AQ with pure DP accounting and Hist. This privacy budget allows a small minimum gap between ranks, thus, allowing us increase the dataset only $6$ times obtaining $293052$ data points. Plots $a$ and $b$ are for substitute adjacency, while $c$ and $d$ correspond to add/remove adjacency.

Figure 4: Comparison of SliceQuantiles, AQ and Hist.

