# OpenReview forum: "Differentially Private Quantiles with Smaller Error"
_NeurIPS.cc/2025/Conference — NeurIPS 2025 poster_

### Official Review · Reviewer_TKv1 · 2025-06-23

**Clarity:** 2
**Significance:** 3
**Originality:** 3
**Rating:** 4
**Confidence:** 3

**Summary:**

The paper presents a novel mechanism for computing approximate quantiles under $\epsilon$-differential privacy (DP) and $(\epsilon, \delta)$-DP, achieving improved error bounds compared to prior work, notably Kaplan et al. (ICML '22). The authors leverage continual counting techniques to reduce the maximum rank error, particularly when the number of quantiles $m$ is large. The paper also includes experimental validation, showing practical improvements.

**Questions:**

1. Minimum Gap Requirement:
   The minimum gap assumption, $\left|q_i - q_{i-1}\right| \geq \Omega\left(\frac{m \log(m) \log(b)}{n \varepsilon}\right)$, is a critical condition for the $\varepsilon$-DP mechanism but differs in nature from Kaplan et al.'s requirement, which pertains to the spacing of data points. Here, the constraint is on the quantiles’ orders, which may limit applicability in certain scenarios. Could the authors elaborate on the implications of this assumption compared to Kaplan et al.'s, particularly regarding its impact on practical settings and datasets where quantiles may be closely spaced?

2. Clarity of Presentation:
   The exposition of the proposed solution could be improved for better accessibility. I suggest adding a sentence clarifying that, under suitable assumptions, $x_{(nq_i)}$ can be approximated by applying a location algorithm (e.g., a private median algorithm) to a subset of points around the quantile. The approach can be described as decoupling into two steps: (1) splitting the dataset into disjoint subsets around the quantiles using private continual counting algorithms, and (2) applying a location algorithm to each subset. This would provide a clearer high-level intuition for the SliceQuantiles algorithm.

3. Potential Bias in Location Algorithm:
   The method relies on estimating the location of data slices around quantiles. If the location algorithm used in SingleQuantile is not a median estimator (e.g., a mean estimator), this could introduce bias in the quantile estimates, unlike Kaplan et al.'s approach, which may be more robust in certain cases. Could the authors comment on whether this is a concern and how the choice of location algorithm affects bias, particularly for non-median estimators?

4. Empirical CDF/Quantile Function Plots:
   To better understand the practical improvements and potential biases, I recommend including plots of the empirical cumulative distribution functions or quantile functions for the datasets used in the experiments. These plots would help readers assess the distributions where the proposed method outperforms Kaplan et al. and whether the generic location algorithm in the $(\varepsilon, \delta)$-DP setting introduces bias, especially when the quantile function has high curvature.

5. Literature:
   The paper references Lalanne et al. (2023), noting that their algorithm relies on strong smoothness assumptions for randomized quantiles. I believe this refers to their histogram estimator for density, used as a plug-in for the quantile function. While Lalanne et al.'s approach requires regularity assumptions, the current paper assumes a minimum gap of 1 between data points, which is a form of regularity under which it should be possible to study histogram estimators. I suggest including a naive baseline similar to Lalanne et al.'s histogram estimator in the experiments, acknowledging the challenge of choosing the bandwidth parameter. This would provide a more comprehensive comparison and highlight the trade-offs of different assumptions.

6. Notation and Clarity Issues:
   - In line 124, the ordered statistics notation ($x_{(i)}$) is used without prior introduction, which could confuse readers unfamiliar with the convention. I recommend defining this notation explicitly.
   - In Equation (1), the use of $d$ for a quantity in $\{-1, 1\}$ is potentially misleading, as $d$ is commonly used to denote dimension. Consider using a different symbol to avoid confusion.
   - In Lemma 3.1, the equation appears to have a typo: it should likely be $\Pr[\mathrm{CC}(\mathbf{r}) = \tilde{\mathbf{r}}] \leq e^\varepsilon \Pr[\mathrm{CC}(\mathbf{r} + \mathbf{e}) = \tilde{\mathbf{r}}]$ instead of comparing $\mathrm{CC}(\mathbf{r})$ to itself. Additionally, in the following line, $\tilde{\mathbf{r}}$ should be replaced with $\mathrm{CC}(\mathbf{r})$ for consistency.
   - In lines 154–156, the bounds for applying SingleQuantile are unclear. Initially, it seems the algorithm uses the upper and lower data points as interval bounds, which could raise privacy concerns. However, it later appears that the bounds are the entire base interval $(a, b)$. Clarifying this and discussing how the domain reduction strategy compares to Kaplan et al.'s tree-based approach would improve understanding.
   - In Lemma 4.1, the use of $F^+$ and $F^-$ for the rectifying functions is confusing, as these symbols are often associated with CDFs or quantile functions in this context. Consider alternative notation to avoid ambiguity.

7. Scalability and Limitations:
   The minimum gap requirement scaling as $\frac{m}{n \varepsilon}$ for $\varepsilon$-DP makes the results less applicable in the high-quantiles regime. For $(\varepsilon, \delta)$-DP, the linear scaling of $\delta$ with $m$ int the $\delta$ part might be concerning, and the claim of utility bounds being independent of $\delta$ suggests this dependence may be hidden in the gap assumptions. Could the authors clarify how this dependence is managed and discuss the practicality of the gap assumptions in high-quantiles scenarios?

**Ethical Concerns:**

["NO or VERY MINOR ethics concerns only"]

**Final Justification:**

After reading the author’s response, I’ve decided to keep my positive rating of the article.

**Limitations:**

Yes

**Quality:**

3

**Strengths And Weaknesses:**

Strengths:
- Improved Error Bounds: The proposed mechanism achieves a maximum rank error of $O\left(\frac{\log(b) + \log^2(m)}{\epsilon}\right)$ for $\epsilon$-DP, improving over Kaplan et al.'s $O\left(\frac{\log(b)\log^2(m)}{\epsilon}\right)$ by a factor of $\Omega\left(\min(\log(b), \log^2(m))\right)$. This is a meaningful theoretical advancement, especially for large $m$.
- Novel Approach: The use of continual counting to randomize quantiles in a correlated manner is innovative and distinguishes this work from prior divide-and-conquer strategies.
- Experimental Validation: The experiments confirm the theoretical improvements, showing a factor of 2 improvement in accuracy over Kaplan et al. when estimating 200 quantiles with $\epsilon=1$ and $\delta=10^{-16}$ under substitute adjacency.
- $(\epsilon, \delta)$-DP Mechanism: The relaxation of the gap assumption in the $(\epsilon, \delta)$-DP mechanism, with error independent of $\delta$, is a notable contribution, particularly when $\delta$ is small. I have questions on this part that are detailed below.

Weaknesses:
See my questions below.

---

> ### Author Rebuttal · Authors · 2025-07-30
>
> Thanks for your questions and your valuable feedback. We structured our rebuttal by first providing a common response, followed by answers to specific questions.
>
> ## Common Response
>
> ### Justification
> Our algorithms are designed to produce very accurate estimates for $m$ input quantiles, where $m$ is sufficiently small compared to $n$. This setting is practical and common—data analysts often care about a limited number of summary statistics (e.g., 10\%, 20\%, ..., 90\%).
> A particularly important case is when the quantiles are equally spaced, which is closely tied to the problem of CDF estimation, the subject of a long line of research (see related work section).  For approximate DP, our gap assumption is relatively mild, at $O(\frac{1}{n \varepsilon}(\log(b) + \log(m) \log \frac{m}{\delta}))$---both terms are typically significantly less than $n$ in practice.
> For equally-spaced quantiles, this means that $\Omega(\frac{n}{\log n \log(n/\delta)})$ quantiles can be answered. For example, with $n = 500,000$ and reasonable choices of $\varepsilon, b, m, \delta$, this gap assumption worked out to less than 0.005 in our experiments, meaning that up to 200 equally-spaced quantiles could be answered. For pure DP, our gap assumption grows slightly as $O(\frac{1}{n \varepsilon} m \log(b) \log (m))$, but remains manageable as long as  $\frac{m}{n} \ll \varepsilon \log (b) \log(m)$. This still permits answering $\Omega(\sqrt{\frac{n \varepsilon}{\log(b) \log(n)}})$ equally-spaced quantiles. Finally, note that our gap assumption can be eliminated altogether by merging overly close quantiles into a single representative quantile; this will then add the required gap to the error instead.
>
> ### Comparison to prior work
>  The main comparison algorithms are that of Kaplan (2022) and Cohen (2023). For data domains which are not extremely large (such as $b = 2^{64})$, the algorithm of Kaplan (2022) is the prior state-of-the-art: it satisfies pure DP and makes no assumption on the quantile gap. However, if the quantiles have the spacing of the prior paragraph (for pure DP), then our algorithm enjoys lower error by a \min\{log^2(m), \log(b)\} factor. If one is willing to relax to approximate DP, our algorithm still has the lower error, and the gap assumption is significantly reduced. By combining quantiles, the gap assumption can be eliminated entirely, and the error will be $O(\frac{1}{\varepsilon}(\log(b) + \log(m) \log \frac{m}{\delta}))$; this is still less than Kaplan (2022) when $\frac{1}{\delta} \ll b$, usually the case for larger data domains.
> For extremely large data domains, the algorithm of Cohen (2023) can start to outperform our algorithm using more advanced techniques than the exponential mechanism. Specifically, their algorithm can be used to answer any quantiles with error $\frac{1}{n \varepsilon} \log^*b \log^2(\frac{1}{\delta})$. However, for domain sizes encountered in practice, the $\log^2(\frac{1}{\delta})$ will likely outweigh this improvement.
> We will add the above discussion in a revised version.
>
> ## Answering Specific Questions
> > 1. Minimum Gap Requirement: ...
>
> We stress that our gap quantile assumption refers to spacing between the input (queried) quantiles, and does not refer to any part of the data itself. Kaplan et al. make an assumption that each point in the dataset itself is separated from the others by a small gap, and enforce this in practice by adding a small amount of random noise to each point. We also do this. For a full discussion of our quantile gap assumption, see the common response.
>
> > 2. Clarity of Presentation: ...
>
> We thank the reviewer for the suggestion and we agree that this approach will enhance the intuition around our algorithm. We will clarify this in a revised version.
>
> > 3. Potential Bias in Location Algorithm: ...
>
> The location algorithm can be thought of as an *interior point* algorithm which aims at returning any data point inside a given range. Any pure DP interior point algorithm can be used as a building block of SliceQuantiles. However, it is necessary to carefully compute the length $2h$ of the slices so that no interior points algorithm fails. The choice of using the exponential mechanism to estimate the median of each slice was driven by two factors: it matches in utility a lower bound of $\Omega(\log b)$ for the interior problem under pure DP [1], and it solves it with high probability thus allowing to set $h = \Omega(\log(b) + \log(m))$. Nevertheless, any pure DP interior point mechanism that achieves rank error $O(h)$ with high probability in $m$ can be used.
>
> [1]: *Mark Bun, Kobbi Nissim, Uri Stemmer, and Salil Vadhan. Differentially private release and learning of threshold functions.*
>
> > 4. Empirical CDF/Quantile Function Plots: ...
>
> We agree that adding plots of the dataset CDFs will be interesting to include. While our theoretical analysis applies to any interior point algorithm, experimentally we always apply the exponential mechanism, even for approximate DP; thus, we do not expect a significant difference in the empirical bias of our algorithm vs. that of Kaplan et al. We will consider adding plots of this in a later version.
>
> > 5. Literature: ...
>
> Thank you for raising these points, which we will discuss more thoroughly in the revision. As pointed out earlier, our gap assumption is an assumption about the query, not the data. The assumption of having a minimum gap of 1 in the data is made for simplicity and convenience to be able to state clear results that are directly comparable between the cases of discrete and continuous domains. The assumption can be enforced whp. by scaling the data and adding random noise to data values, but this makes the utility guarantee more complicated to state.
>
> Concerning a comparison to Lalanne et al. (2023) we agree that it would be a good idea to include it as a baseline. Theoretically speaking, their method is vulnerable to situations in which the distribution of values is concentrated (or concentrated around a few values) in which case even the non-private histogram may not contain enough information to output good quantile estimates.
>
> > 6. Notation and Clarity Issues:
>
> Thank you for pointing out these issues, we will make sure to fix them.
>
>
> > 7. Scalability and Limitations:
>
> We show that approximate DP allows us to achieve the same error as we were able to show for pure DP with a weaker gap assumption than was required for pure DP, since the quantile gap requirement scales with $\log(m/\delta)$. Therefore, increasing $\delta$ will decrease the quantile gap required for good utility and make our results more applicable in higher-quantile regimes. It is correct that the utility bounds are independent of $\delta$, and $\delta$ only affects the gap assumption. See our common response for a full discussion of this interaction.

---

> > ### Comment · Reviewer_TKv1 · 2025-08-04
> >
> > I would like to express my gratitude to the authors for their insightful response, which effectively addressed the queries I had regarding the article. Consequently, I will retain my positive rating for the article.

---

### Official Review · Reviewer_SjbT · 2025-07-02

**Clarity:** 3
**Significance:** 3
**Originality:** 2
**Rating:** 5
**Confidence:** 3

**Summary:**

This paper introduces SliceQuantiles, a new mechanism for privacy preserving quantile approximation. While the existing works, in $\epsilon$ $\delta$ DP are optimal in logb, the proposed method improves over the state-of-the-art in reducing the complexity of $\delta$ term (error independent of $\delta$). In pure DP setting, the proposed method saves min(logb, log^2m) factor. The improvement is achieved by combining continual counting techniques with the exponential mechanism. Using Adult data set, the authors demonstrate that the proposed SliceQuantiles outperform AQ.

**Questions:**

Please see W1 - W3

**Ethical Concerns:**

["NO or VERY MINOR ethics concerns only"]

**Final Justification:**

Having reviewed the authors’ response for all reviewers, I have decided to maintain my positive score.

**Limitations:**

yes

**Paper Formatting Concerns:**

looks good to me

**Quality:**

3

**Strengths And Weaknesses:**

S1.  The use of continual counting to correlate quantile noise is clever. The main idea is sound and clearly presented.

S2. The paper provides rigorous theoretical proofs for both privacy and utility guarantees under add/remove model for approximate/pure DP. (Didn't check the proofs in detail)

S3. Using scaled Adult dataset, authors demonstrated the high accuracy of the proposed algorithm.

W1. For clarity: please consider to i) add input dataset in algorithm 1, ii) add ticks in Figure 1, iii) compare with AppindExp (reference 12), and iv) include approximate quantile literatures in the streaming model such as `Differentially private linear sketches: Efficient implementations and applications`, `Improved utility analysis of private countsketch`, and `Bounded space differentially private quantiles`.

W2. Need more discussions or experiment on the query time complexity (perhaps appendix).

W3. What happens to the utility guarantee under substitute model?

---

> ### Author Rebuttal · Authors · 2025-07-30
>
> Thanks for your interest in our paper! We would like to answer your specific questions.
>
> > For clarity: please consider to i) add input dataset in algorithm 1, ii) add ticks in Figure 1, iii) compare with AppindExp (reference 12), and iv) include approximate quantile literatures in the streaming model such as Differentially private linear sketches: Efficient implementations and applications, Improved utility analysis of private countsketch, and Bounded space differentially private quantiles.
>
> Thanks for this suggestion and we will incorporate them in the revised version. We chose to compare with Kaplan et al as our baseline since it is the state-of-the-art private quantile estimation algorithm in the central model and outperforms [12] (as validated via experiments in their paper).
>
> > Need more discussions or experiment on the query time complexity (perhaps appendix).
>
> The running time of the algorithm is $O(n \log n)$, it is dominated by the time taken to sort the dataset. Experimentally, we were able to run our algorithm with 200 quantiles and 500,000 data points in under a second.
>
> > What happens to the utility guarantee under substitute model?
>
> As the utility of our algorithm does not depend on $\delta$, the utility guarantee under substitution analysis is affected only by a factor of 2 in the privacy budget $\varepsilon$ (Corollary 5.2, 5.3 with privacy budget given in Corollary 4.4).

---

### Official Review · Reviewer_Ky2d · 2025-07-02

**Clarity:** 2
**Significance:** 3
**Originality:** 4
**Rating:** 4
**Confidence:** 4

**Summary:**

This paper improved error bound in the problem of estimating multiple quantiles of a dataset under DP over prior methods. The authors introduce the SliceQuantiles mechanism, which offers both pure ε-DP and approximate (ε, δ)-DP guarantees. Their central innovation lies in leveraging continual counting techniques to inject correlated noise into the rank estimates of quantiles, enabling more efficient privacy composition.

**Questions:**

Could the authors provide a clearer and earlier definition of what constitutes a "slice" and explain explicitly why a slice corresponds to a subproblem in their framework?

The proposed setting relies on a minimal gap between quantiles, but verifying that this condition holds may be non-trivial. Can the authors provide guidance or practical procedures for how one might check or enforce this minimal gap in realistic applications? Also, what is the motivation behind the rank error metric?

The performance deteriorates for larger $b$ and depend on minimal gap. How does $b$ and gap interacts? It appear to me that a larger minimal gap leads to a larger $b$.

**Ethical Concerns:**

["NO or VERY MINOR ethics concerns only"]

**Final Justification:**

The authors have clarified the connection of this work with prior work and have addressed the assumption regarding the minimum gap. The main concern remains the clarity of the presentation and the lack of clearer definitions for key concepts (e.g., “slices”). I assume these issues can be addressed in the final version.

**Limitations:**

yes

**Quality:**

3

**Strengths And Weaknesses:**

Strengths:
The paper presents a theoretically sound improvement over prior work, eliminating a factor in the error bound.

The use of continual counting for quantile estimation is original in this context.

Weaknesses:

This paper is difficult to follow and understand in several key aspects. For example, the concept of a "slice" is introduced at line 75 as “m subproblems, referred to as ‘slices’.” This term is referenced multiple times before line 124, where it is only hinted that a slice contains a subset of the dataset $X$. A more concrete definition appears at line 130, where a slice is described as some subset of $X$ determined by a sufficiently large $h$; however, it is not made clear what constitutes a "sufficiently large" $h$. From that point onward, slices are treated as subsets of $X$ in various contexts, but it remains unclear why a slice corresponds to a subproblem in the proposed methodology.

Additionally, in line 185, the Technical Challenge section attempts to explain the difficulty of establishing the differential privacy condition by describing a failed approach. However, the exposition of this failed attempt is unclear, and it does not effectively elucidate the core technical challenge faced by the authors.

While supported by prior literature, the problem formulation appears artificial and narrow. The paper focuses on controlling the rank error under the condition that queried quantiles are separated by a minimal gap. Although it is reasonable that rank error becomes unavoidable when quantiles are too close, the authors do not provide sufficient justification for the practical relevance of this setting, especially given that verifying the minimal quantile gap condition may be non-trivial in practice which narrow down the applications by a lot.

---

> ### Author Rebuttal · Authors · 2025-07-30
>
> Thanks for your questions and your consideration of our paper. We structured our rebuttal by first providing a common response, followed by answers to specific questions.
>
> ## Common Response
>
> ### Justification
> Our algorithms are designed to produce very accurate estimates for $m$ input quantiles, where $m$ is sufficiently small compared to $n$. This setting is practical and common—data analysts often care about a limited number of summary statistics (e.g., 10\%, 20\%, ..., 90\%).
> A particularly important case is when the quantiles are equally spaced, which is closely tied to the problem of CDF estimation, the subject of a long line of research (see related work section).  For approximate DP, our gap assumption is relatively mild, at $O(\frac{1}{n \varepsilon}(\log(b) + \log(m) \log \frac{m}{\delta}))$---both terms are typically significantly less than $n$ in practice.
> For equally-spaced quantiles, this means that $\Omega(\frac{n}{\log n \log(n/\delta)})$ quantiles can be answered. For example, with $n = 500,000$ and reasonable choices of $\varepsilon, b, m, \delta$, this gap assumption worked out to less than 0.005 in our experiments, meaning that up to 200 equally-spaced quantiles could be answered. For pure DP, our gap assumption grows slightly as $O(\frac{1}{n \varepsilon} m \log(b) \log (m))$, but remains manageable as long as  $\frac{m}{n} \ll \varepsilon \log (b) \log(m)$. This still permits answering $\Omega(\sqrt{\frac{n \varepsilon}{\log(b) \log(n)}})$ equally-spaced quantiles. Finally, note that our gap assumption can be eliminated altogether by merging overly close quantiles into a single representative quantile; this will then add the required gap to the error instead.
>
> ### Comparison to prior work
>  The main comparison algorithms are that of Kaplan (2022) and Cohen (2023). For data domains which are not extremely large (such as $b = 2^{64})$, the algorithm of Kaplan (2022) is the prior state-of-the-art: it satisfies pure DP and makes no assumption on the quantile gap. However, if the quantiles have the spacing of the prior paragraph (for pure DP), then our algorithm enjoys lower error by a \min\{log^2(m), \log(b)\} factor. If one is willing to relax to approximate DP, our algorithm still has the lower error, and the gap assumption is significantly reduced. By combining quantiles, the gap assumption can be eliminated entirely, and the error will be $O(\frac{1}{\varepsilon}(\log(b) + \log(m) \log \frac{m}{\delta}))$; this is still less than Kaplan (2022) when $\frac{1}{\delta} \ll b$, usually the case for larger data domains.
> For extremely large data domains, the algorithm of Cohen (2023) can start to outperform our algorithm using more advanced techniques than the exponential mechanism. Specifically, their algorithm can be used to answer any quantiles with error $\frac{1}{n \varepsilon} \log^*b \log^2(\frac{1}{\delta})$. However, for domain sizes encountered in practice, the $\log^2(\frac{1}{\delta})$ will likely outweigh this improvement.
> We will add the above discussion in a revised version.
>
> ## Answering Specific Questions
> > the Technical Challenge section attempts to explain the difficulty of establishing the differential privacy condition by describing a failed approach. However, the exposition of this failed attempt is unclear, and it does not effectively elucidate the core technical challenge faced by the authors.
>
>  Briefly, our algorithm works by sorting the dataset and taking a contiguous subset of data around the desired quantile. We refer to each contiguous subset of data as a “slice”, and this is what is used to form a private estimate of the quantile. The intuition behind this approach is that the data points that are closest to the queried quantile should be used to privately estimate the quantile as accurately as possible.
>
> The technical challenge of this approach is that, perhaps surprisingly, the above procedure is not private because modifying one data point can modify the data contained in many slices (see lines 126-134 for an example). To circumvent this issue, we add correlated noise to each quantile before forming the slices which has the effect of *hiding* the modifications created in all the slices by the change in a single data point. We will clarify this in a revised version.
>
> > Could the authors provide a clearer and earlier definition of what constitutes a "slice" and explain explicitly why a slice corresponds to a subproblem in their framework?
>
> In summary, a part of the definition can be found in line 124 where we say that a slice is “a contiguous subsequence of the sorted input data”. Additionally, the slice must be centered on a specific rank. We will make sure to introduce this concept earlier in the paper and to define it properly.
>
> > The proposed setting relies on a minimal gap between quantiles, but verifying that this condition holds may be non-trivial. Can the authors provide guidance or practical procedures for how one might check or enforce this minimal gap in realistic applications? Also, what is the motivation behind the rank error metric?
>
> We stress that the minimal gap assumption is an assumption about the quantiles being queried and can be easily checked by computing the differences between consecutive quantiles (which is publicly known). In particular, it does not rely on any assumption about the data. See the common response for a discussion of the gap assumption.
> The rank error metric is natural since it measures how far off an approximate quantile is from having the desired number of data points on each side. We note that under differential privacy it is impossible in general to achieve approximation of the value of a quantile because the value does not have small sensitivity. However, assuming that the data distribution is reasonably smooth, a small rank error implies a small error in value, too. For these reasons, rank error is the most common error measure for private quantile estimates and is the standard metric used in existing literature, e.g:
>
> *Gillenwater, Jennifer et al. “Differentially Private Quantiles.” ArXiv abs/2102.08244 (2021): n. pag.*
>
> *Kaplan, H., Schnapp, S. &amp; Stemmer, U.. (2022). Differentially Private Approximate Quantiles. Proceedings of the 39th International Conference on Machine Learning.*
>
> *Bassily, Raef, and Adam Smith. "Local, private, efficient protocols for succinct histograms." Proceedings of the forty-seventh annual ACM symposium on Theory of computing. 2015.*
>
> > The performance deteriorates for larger $b$  and depend on minimal gap. How does $b$ and gap interacts? It appear to me that a larger minimal gap leads to a larger $b$.
>
> Existing lower bounds show that the rank error must grow with $b$, both under pure and approximate differential privacy. The gap we need between quantiles is proportional to $\log(b)$. Since the gap is inversely proportional to the number of data points $n$, the value of $n$ needed to answer a given set of quantile queries grows proportionally with $\log(b)$.

---

> > ### Comment · Reviewer_Ky2d · 2025-08-06
> >
> > Thank you to the authors for the response, which addressed most of my concerns. My remaining reservation pertains to the clarity and self-contained nature of the paper. However, I trust that the authors will resolve these issues in the camera-ready version. Accordingly, I will revise my score to 4 after the discussion period.

---

> ### Author Response · Authors · 2025-08-05
> **Follow-up questions**
>
> Dear reviewer,
>
> We are happy to answer any follow-up questions and comments that you might have. Thanks for your time and feedback!

---

### Official Review · Reviewer_v3P9 · 2025-07-03

**Clarity:** 4
**Significance:** 2
**Originality:** 3
**Rating:** 4
**Confidence:** 4

**Summary:**

This paper proposes a novel differentially private algorithm for the quantile estimation problem, improving upon prior utility guarantees. Specifically, the authors reduce the approximation error from the previous $O(\log(b)\log^2(m)/\epsilon)$ to $O((\log(b)+\log^2(m))/\epsilon)$. However, to achieve this improvement, the algorithm additionally assumes that the queried quantiles satisfy a spacing condition:
$|q_i-q_{i+1}|\geq\Omega(m\log(m)\log(b)/n\epsilon)$. In the case of $(\epsilon,\delta)$-differential privacy, this constraint is relaxed to
$|q_i-q_{i+1}|\geq\Omega(\log(m)\log(m/\delta)+\log(b)/n\epsilon)$ while maintaining the same utility bound. The paper also presents experimental results on the AdultAge and AdultHours datasets, demonstrating improved performance over previous approaches.

A key technical contribution is the use of a sliced quantile approach, in which the dataset is partitioned (via noisy boundaries) into disjoint slices, and a single-quantile mechanism is applied independently within each slice. The disjointness of slices enables strong utility guarantees through statistical independence.

**Questions:**

In the experiments section, could the authors provide a more detailed analysis of how the algorithm performs under varying values of the privacy parameters $\epsilon$ and $\delta$? This would help assess robustness across different privacy regimes.

**Ethical Concerns:**

["NO or VERY MINOR ethics concerns only"]

**Final Justification:**

The authors' detailed response answered my questions. Since my rating is positive, I will keep my rating.

**Limitations:**

This is a theoretical work, and I do not see any apparent negative societal impacts.

**Paper Formatting Concerns:**

I have no concerns regarding the formatting of the paper.

**Quality:**

3

**Strengths And Weaknesses:**

Strengths:

1.The paper is clearly structured and well written.

2.The utility improvement from $O(\log(b)\log^2(m)/\epsilon)$ to $O((\log(b)+\log^2(m))/\epsilon)$ is substantial and well motivated.

3.The theoretical analysis is sound.

4.Experimental results show meaningful gains over existing methods.

Weaknesses

1.The core idea of slicing queries into independent subproblems is conceptually simple and may be seen as a straightforward extension.

2.The spacing assumption imposed on the quantiles lacks sufficient justification or motivation, and its relationship to prior work is underexplored.

---

> ### Author Rebuttal · Authors · 2025-07-30
>
> Thanks for your questions and your consideration of our paper. We structured our rebuttal by first providing a common response, followed by answers to specific questions.
>
> ## Common Response
>
> ### Justification
> Our algorithms are designed to produce very accurate estimates for $m$ input quantiles, where $m$ is sufficiently small compared to $n$. This setting is practical and common—data analysts often care about a limited number of summary statistics (e.g., 10\%, 20\%, ..., 90\%).
> A particularly important case is when the quantiles are equally spaced, which is closely tied to the problem of CDF estimation, the subject of a long line of research (see related work section).  For approximate DP, our gap assumption is relatively mild, at $O(\frac{1}{n \varepsilon}(\log(b) + \log(m) \log \frac{m}{\delta}))$---both terms are typically significantly less than $n$ in practice.
> For equally-spaced quantiles, this means that $\Omega(\frac{n}{\log n \log(n/\delta)})$ quantiles can be answered. For example, with $n = 500,000$ and reasonable choices of $\varepsilon, b, m, \delta$, this gap assumption worked out to less than 0.005 in our experiments, meaning that up to 200 equally-spaced quantiles could be answered. For pure DP, our gap assumption grows slightly as $O(\frac{1}{n \varepsilon} m \log(b) \log (m))$, but remains manageable as long as  $\frac{m}{n} \ll \varepsilon \log (b) \log(m)$. This still permits answering $\Omega(\sqrt{\frac{n \varepsilon}{\log(b) \log(n)}})$ equally-spaced quantiles. Finally, note that our gap assumption can be eliminated altogether by merging overly close quantiles into a single representative quantile; this will then add the required gap to the error instead.
>
> ### Comparison to prior work
>  The main comparison algorithms are that of Kaplan (2022) and Cohen (2023). For data domains which are not extremely large (such as $b = 2^{64})$, the algorithm of Kaplan (2022) is the prior state-of-the-art: it satisfies pure DP and makes no assumption on the quantile gap. However, if the quantiles have the spacing of the prior paragraph (for pure DP), then our algorithm enjoys lower error by a $\min(\log^2(m), \log(b))$ factor. If one is willing to relax to approximate DP, our algorithm still has the lower error, and the gap assumption is significantly reduced. By combining quantiles, the gap assumption can be eliminated entirely, and the error will be $O(\frac{1}{\varepsilon}(\log(b) + \log(m) \log \frac{m}{\delta}))$; this is still less than Kaplan (2022) when $\frac{1}{\delta} \ll b$, usually the case for larger data domains.
> For extremely large data domains, the algorithm of Cohen (2023) can start to outperform our algorithm using more advanced techniques than the exponential mechanism. Specifically, their algorithm can be used to answer any quantiles with error $\frac{1}{n \varepsilon} \log^*b \log^2(\frac{1}{\delta})$. However, for domain sizes encountered in practice, the $\log^2(\frac{1}{\delta})$ will likely outweigh this improvement.
> We will add the above discussion in a revised version.
>
> ## Answering Specific Questions
> > The core idea of slicing queries into independent subproblems is conceptually simple and may be seen as a straightforward extension.
>
> Note that the queries are *not* treated as independent subproblems, as this is not private (full discussion in Section 3.1). Instead, our algorithm adds correlated noise to shift each quantile, and the privacy analysis only goes through on this global structure.
>
> > The spacing assumption imposed on the quantiles lacks sufficient justification or motivation, and its relationship to prior work is underexplored.
>
> Please refer to our common response for more discussion on this. We will add this clarification in a revised version.
>
> > In the experiments section, could the authors provide a more detailed analysis of how the algorithm performs under varying values of the privacy parameters  and ? This would help assess robustness across different privacy regimes.
>
>  $\delta$ does not affect the utility of the mechanism, but only the initial assumption, i.e. the smaller the $\delta$ the larger the quantile gap must be.
> We include some additional results for more values of $\varepsilon$. The observation is similar to the results provided in the main paper – our algorithm outperforms Kaplan et al. by a factor 2 and a factor 1.3 under substitution privacy for $\varepsilon = 0.5$ and $\varepsilon= 5$, respectively.

---

> > ### Comment · Reviewer_v3P9 · 2025-08-05
> >
> > Thank authors for the response. I will keep my score as it is.

---

### Decision · Program_Chairs · 2025-09-17

**Decision:**

Accept (poster)

**Comment:**

The paper provides new pure and approximate DP versions of a new method for estimating multiple quantiles by applying techniques from the continual counting literature.

Reviewers appreciated the novelty of the paper's methods and improved error guarantees. The most pressing recurring concern was over clarity, though this was at least partially addressed through author-reviewer discussions. The authors should make sure to revisit these discussions and use them to clarify the final version of the paper.